# Randomized Block Krylov Methods for Stronger and Faster Approximate Singular Value Decomposition

**Cameron Musco**
Massachusetts Institute of Technology, EECS
Cambridge, MA 02139, USA
cnmusco@mit.edu

**Christopher Musco**
Massachusetts Institute of Technology, EECS
Cambridge, MA 02139, USA
cpmusco@mit.edu

## Abstract

Since being analyzed by Rokhlin, Szlam, and Tygert [1] and popularized by Halko, Martinsson, and Tropp [2], randomized Simultaneous Power Iteration has become the method of choice for approximate singular value decomposition. It is more accurate than simpler sketching algorithms, yet still converges quickly for *any* matrix, independently of singular value gaps. After $\tilde{O}(1/\epsilon)$ iterations, it gives a low-rank approximation within $(1 + \epsilon)$ of optimal for spectral norm error.

We give the first provable runtime improvement on Simultaneous Iteration: a randomized block Krylov method, closely related to the classic Block Lanczos algorithm, gives the same guarantees in just $\tilde{O}(1/\sqrt{\epsilon})$ iterations and performs substantially better experimentally. Our analysis is the first of a Krylov subspace method that does not depend on singular value gaps, which are unreliable in practice.

Furthermore, while it is a simple accuracy benchmark, even $(1 + \epsilon)$ error for spectral norm low-rank approximation does not imply that an algorithm returns high quality principal components, a major issue for data applications. We address this problem for the first time by showing that both Block Krylov Iteration and Simultaneous Iteration give nearly optimal PCA for any matrix. This result further justifies their strength over non-iterative sketching methods.

## 1 Introduction

Any matrix $\mathbf{A} \in \mathbb{R}^{n \times d}$ with rank $r$ can be written using a singular value decomposition (SVD) as $\mathbf{A} = \mathbf{U}\mathbf{\Sigma}\mathbf{V}^T$. $\mathbf{U} \in \mathbb{R}^{n \times r}$ and $\mathbf{V} \in \mathbb{R}^{d \times r}$ have orthonormal columns ($\mathbf{A}$'s left and right singular vectors) and $\mathbf{\Sigma} \in \mathbb{R}^{r \times r}$ is a positive diagonal matrix containing $\mathbf{A}$'s singular values: $\sigma_1 \geq \ldots \geq \sigma_r$. A rank $k$ *partial SVD* algorithm returns just the top $k$ left or right singular vectors of $\mathbf{A}$. These are the first $k$ columns of $\mathbf{U}$ or $\mathbf{V}$, denoted $\mathbf{U}_k$ and $\mathbf{V}_k$ respectively.

Among countless applications, the SVD is used for optimal low-rank approximation and principal component analysis (PCA). Specifically, for $k < r$, a partial SVD can be used to construct a rank $k$ approximation $\mathbf{A}_k$ such that both $\|\mathbf{A} - \mathbf{A}_k\|_F$ and $\|\mathbf{A} - \mathbf{A}_k\|_2$ are as small as possible. We simply set $\mathbf{A}_k = \mathbf{U}_k\mathbf{U}_k^T\mathbf{A}$. That is, $\mathbf{A}_k$ is $\mathbf{A}$ projected onto the space spanned by its top $k$ singular vectors.

For principal component analysis, $\mathbf{A}$'s top singular vector $\mathbf{u}_1$ provides a top principal component, which describes the direction of greatest variance within $\mathbf{A}$. The $i^{\text{th}}$ singular vector $\mathbf{u}_i$ provides the $i^{\text{th}}$ principal component, which is the direction of greatest variance orthogonal to all higher principal components. Formally, denoting $\mathbf{A}$'s $i^{\text{th}}$ singular value as $\sigma_i$,

$$\mathbf{u}_i^T\mathbf{A}\mathbf{A}^T\mathbf{u}_i = \sigma_i^2 = \max_{\mathbf{x}:\|\mathbf{x}\|_2=1,\ \mathbf{x}\perp\mathbf{u}_j\forall j<i}\mathbf{x}^T\mathbf{A}\mathbf{A}^T\mathbf{x}.$$

Traditional SVD algorithms are expensive, typically running in $O(nd^2)$ time, so there has been substantial research on randomized techniques that seek nearly optimal low-rank approximation and

PCA [3, 4, 1, 2, 5]. These methods are quickly becoming standard tools in practice and implementations are widely available [6, 7, 8, 9], including in popular learning libraries [10].

Recent work focuses on algorithms whose runtimes *do not depend on properties of* $\mathbf{A}$. In contrast, classical literature typically gives runtime bounds that depend on the gaps between $\mathbf{A}$'s singular values and become useless when these gaps are small (which is often the case in practice – see Section 6). This limitation is due to a focus on how quickly approximate singular vectors converge to the actual singular vectors of $\mathbf{A}$. When two singular vectors have nearly identical values they are difficult to distinguish, so convergence inherently depends on singular value gaps.

Only recently has a shift in approximation goal, along with an improved understanding of randomization, allowed for algorithms that avoid gap dependence and thus run provably fast for *any* matrix. For low-rank approximation and PCA, we only need to find a subspace that captures nearly as much variance as $\mathbf{A}$'s top singular vectors – distinguishing between two close singular values is overkill.

## 1.1  Prior Work

The fastest randomized SVD algorithms [3, 5] run in $O(\mathrm{nnz}(\mathbf{A}))$ time[1], are based on non-iterative sketching methods, and return a rank $k$ matrix $\mathbf{Z}$ with orthonormal columns $\mathbf{z}_1, \ldots, \mathbf{z}_k$ satisfying

$$\text{Frobenius Norm Error:} \qquad \|\mathbf{A} - \mathbf{Z}\mathbf{Z}^T\mathbf{A}\|_F \leq (1+\epsilon)\|\mathbf{A} - \mathbf{A}_k\|_F. \tag{1}$$

Unfortunately, as emphasized in prior work [1, 2, 11, 12], Frobenius norm error is often hopelessly insufficient, *especially* for data analysis and learning applications. When $\mathbf{A}$ has a "heavy-tail" of singular values, which is common for noisy data, $\|\mathbf{A} - \mathbf{A}_k\|_F^2 = \sum_{i>k} \sigma_i^2$ can be huge, potentially much larger than $\mathbf{A}$'s top singular value. This renders (1) meaningless since $\mathbf{Z}$ does not need to align with any large singular vectors to obtain good multiplicative error.

To address this shortcoming, a number of papers target spectral norm low-rank approximation error,

$$\text{Spectral Norm Error:} \qquad \|\mathbf{A} - \mathbf{Z}\mathbf{Z}^T\mathbf{A}\|_2 \leq (1+\epsilon)\|\mathbf{A} - \mathbf{A}_k\|_2, \tag{2}$$

which is intuitively stronger. When looking for a rank $k$ approximation, $\mathbf{A}$'s top $k$ singular vectors are often considered data and the remaining tail is considered noise. A spectral norm guarantee roughly ensures that $\mathbf{Z}\mathbf{Z}^T\mathbf{A}$ recovers $\mathbf{A}$ up to this noise threshold.

A series of work [1, 2, 13, 14, 15] shows that the decades old Simultaneous Power Iteration (also called subspace iteration or orthogonal iteration) implemented with random start vectors, achieves (2) after $\tilde{O}(1/\epsilon)$ iterations. Hence, this method, which was popularized by Halko, Martinsson, and Tropp in [2], has become the randomized SVD algorithm of choice for practitioners [10, 16].

## 2  Our Results

| **Algorithm 1** SIMULTANEOUS ITERATION | **Algorithm 2** BLOCK KRYLOV ITERATION |
| --- | --- |
| **input**: $\mathbf{A} \in \mathbb{R}^{n \times d}$, error $\epsilon \in (0, 1)$, rank $k \leq n, d$ <br> **output**: $\mathbf{Z} \in \mathbb{R}^{n \times k}$ | **input**: $\mathbf{A} \in \mathbb{R}^{n \times d}$, error $\epsilon \in (0, 1)$, rank $k \leq n, d$ <br> **output**: $\mathbf{Z} \in \mathbb{R}^{n \times k}$ |
| 1: $q := \Theta(\frac{\log d}{\epsilon})$, $\mathbf{\Pi} \sim \mathcal{N}(0,1)^{d \times k}$ <br> 2: $\mathbf{K} := (\mathbf{A}\mathbf{A}^T)^q \mathbf{A}\mathbf{\Pi}$ <br> 3: Orthonormalize the columns of $\mathbf{K}$ to obtain $\mathbf{Q} \in \mathbb{R}^{n \times k}$. <br> 4: Compute $\mathbf{M} := \mathbf{Q}^T\mathbf{A}\mathbf{A}^T\mathbf{Q} \in \mathbb{R}^{k \times k}$. <br> 5: Set $\bar{\mathbf{U}}_k$ to the top $k$ singular vectors of $\mathbf{M}$. <br> 6: **return** $\mathbf{Z} = \mathbf{Q}\bar{\mathbf{U}}_\mathbf{k}$. | 1: $q := \Theta(\frac{\log d}{\sqrt{\epsilon}})$, $\mathbf{\Pi} \sim \mathcal{N}(0,1)^{d \times k}$ <br> 2: $\mathbf{K} := \left[\mathbf{A}\mathbf{\Pi}, (\mathbf{A}\mathbf{A}^T)\mathbf{A}\mathbf{\Pi}, ..., (\mathbf{A}\mathbf{A}^T)^q\mathbf{A}\mathbf{\Pi}\right]$ <br> 3: Orthonormalize the columns of $\mathbf{K}$ to obtain $\mathbf{Q} \in \mathbb{R}^{n \times qk}$. <br> 4: Compute $\mathbf{M} := \mathbf{Q}^T\mathbf{A}\mathbf{A}^T\mathbf{Q} \in \mathbb{R}^{qk \times qk}$. <br> 5: Set $\bar{\mathbf{U}}_k$ to the top $k$ singular vectors of $\mathbf{M}$. <br> 6: **return** $\mathbf{Z} = \mathbf{Q}\bar{\mathbf{U}}_\mathbf{k}$. |

## 2.1  Faster Algorithm

We show that Algorithm 2, a randomized relative of the Block Lanczos algorithm [17, 18], which we call Block Krylov Iteration, gives the same guarantees as Simultaneous Iteration (Algorithm 1) in just $\tilde{O}(1/\sqrt{\epsilon})$ iterations. This not only gives the fastest known theoretical runtime for achieving (2), but also yields substantially better performance in practice (see Section 6).

Even though the algorithm has been discussed and tested for potential improvement over Simultaneous Iteration [1, 19, 20], theoretical bounds for Krylov subspace and Lanczos methods are much more limited. As highlighted in [11],

> "Despite decades of research on Lanczos methods, the theory for [randomized power iteration] is more complete and provides strong guarantees of excellent accuracy, whether or not there exist any gaps between the singular values."

Our work addresses this issue, giving the first gap independent bound for a Krylov subspace method.

## 2.2 Stronger Guarantees

In addition to runtime improvements, we target a much stronger notion of approximate SVD that is needed for many applications, but for which no gap-independent analysis was known.

Specifically, as noted in [21], while intuitively stronger than Frobenius norm error, $(1 + \epsilon)$ spectral norm low-rank approximation error does not guarantee any accuracy in $\mathbf{Z}$ for many matrices[2]. Consider $\mathbf{A}$ with its top $k + 1$ squared singular values all equal to 10 followed by a tail of smaller singular values (e.g. $1000k$ at 1). $\|\mathbf{A} - \mathbf{A}_k\|_2^2 = 10$ but in fact $\|\mathbf{A} - \mathbf{Z}\mathbf{Z}^T\mathbf{A}\|_2^2 = 10$ for *any* rank $k$ $\mathbf{Z}$, leaving the spectral norm bound useless. At the same time, $\|\mathbf{A} - \mathbf{A}_k\|_F^2$ is large, so Frobenius error is meaningless as well. For example, *any* $\mathbf{Z}$ obtains $\|\mathbf{A} - \mathbf{Z}\mathbf{Z}^T\mathbf{A}\|_F^2 \leq (1.01)\|\mathbf{A} - \mathbf{A}_k\|_F^2$.

With this scenario in mind, it is unsurprising that low-rank approximation guarantees fail as an accuracy measure in practice. We ran a standard sketch-and-solve approximate SVD algorithm (see Section 3) on SNAP/AMAZON0302, an Amazon product co-purchasing dataset [22, 23], and achieved very good low-rank approximation error in both norms for $k = 30$:

$$\|\mathbf{A} - \mathbf{Z}\mathbf{Z}^T\mathbf{A}\|_F < 1.001\|\mathbf{A} - \mathbf{A}_k\|_F \quad \text{and} \quad \|\mathbf{A} - \mathbf{Z}\mathbf{Z}^T\mathbf{A}\|_2 < 1.038\|\mathbf{A} - \mathbf{A}_k\|_2.$$

However, the approximate principal components given by $\mathbf{Z}$ are of significantly lower quality than $\mathbf{A}$'s true singular vectors (see Figure 1). We saw similar results for a number of other datasets.

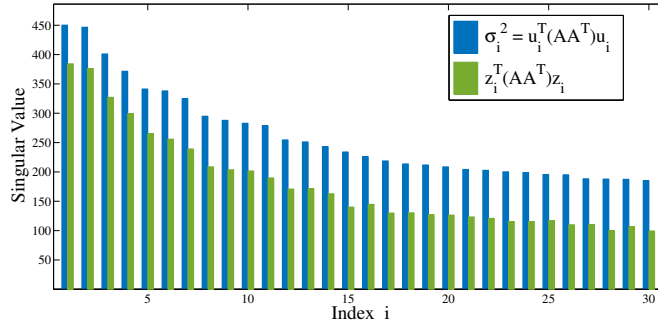

Figure 1: Poor per vector error (3) for SNAP/AMAZON0302 returned by a sketch-and-solve approximate SVD that gives very good low-rank approximation in both spectral and Frobenius norm.

We address this issue by introducing a per vector guarantee that requires each approximate singular vector $\mathbf{z}_1, \ldots, \mathbf{z}_k$ to capture nearly as much variance as the corresponding true singular vector:

$$\text{Per Vector Error:} \qquad \forall i, \ \left| \mathbf{u}_i^T\mathbf{A}\mathbf{A}^T\mathbf{u}_i - \mathbf{z}_i^T\mathbf{A}\mathbf{A}^T\mathbf{z}_i \right| \leq \epsilon\sigma_{k+1}^2. \tag{3}$$

The error bound (3) is very strong in that it depends on $\epsilon\sigma_{k+1}^2$, which is better then relative error for $\mathbf{A}$'s large singular values. While it is reminiscent of the bounds sought in classical numerical analysis [24], we stress that (3) does not require each $\mathbf{z}_i$ to converge to $\mathbf{u}_i$ in the presence of small singular value gaps. In fact, we show that both randomized Block Krylov Iteration and our slightly modified Simultaneous Iteration algorithm achieve (3) in gap-independent runtimes.

## 2.3 Main Result

Our contributions are summarized in Theorem 1. Its detailed proof is relegated to the full version of this paper [25]. The runtimes are given in Theorems 6 and 7, and the three error bounds shown in Theorems 10, 11, and 12. In Section 4 we provide a sketch of the main ideas behind the result.

**Theorem 1** (Main Theorem). *With high probability, Algorithms 1 and 2 find approximate singular vectors $\mathbf{Z} = [\mathbf{z}_1, \ldots, \mathbf{z}_k]$ satisfying guarantees (1) and (2) for low-rank approximation and (3) for PCA. For error $\epsilon$, Algorithm 1 requires $q = O(\log d/\epsilon)$ iterations while Algorithm 2 requires $q = O(\log d/\sqrt{\epsilon})$ iterations. Excluding lower order terms, both algorithms run in time $O(\mathrm{nnz}(\mathbf{A})kq)$.*

In the full version of this paper we also use our results to give an alternative analysis that *does* depend on singular value gaps and can offer significantly faster convergence when $\mathbf{A}$ has decaying singular values. It is possible to take further advantage of this result by running Algorithms 1 and 2 with a $\mathbf{\Pi}$ that has $> k$ columns, a simple modification for accelerating either method.

In Section 6 we test both algorithms on a number of large datasets. We justify the importance of gap independent bounds for predicting algorithm convergence and we show that Block Krylov Iteration in fact significantly outperforms the more popular Simultaneous Iteration.

### 2.4 Comparison to Classical Bounds

Decades of work has produced a variety of gap *dependent* bounds for Krylov methods [26]. Most relevant to our work are bounds for block Krylov methods with block size equal to $k$ [27]. Roughly speaking, with randomized initialization, these results offer guarantees equivalent to our strong equation (3) for the top $k$ singular directions after $O(\log(d/\epsilon)/\sqrt{\sigma_k/\sigma_{k+1} - 1})$ iterations.

This bound is recovered in Section 7 of this paper's full version [25]. When the target accuracy $\epsilon$ is smaller than the relative singular value gap $(\sigma_k/\sigma_{k+1} - 1)$, it is tighter than our gap independent results. However, as discussed in Section 6, for high dimensional data problems where $\epsilon$ is set far above machine precision, gap independent bounds more accurately predict required iteration count.

Prior work also attempts to analyze algorithms with block size *smaller* than $k$ [24]. While "small block" algorithms offer runtime advantages, it is well understood that with $b$ duplicate singular values, it is impossible to recover the top $k$ singular directions with a block of size $< b$ [28]. More generally, large singular value clusters slow convergence, so any small block algorithm must have runtime dependence on the gaps between *each adjacent pair of top singular values* [29].

## 3 Analyzing Simultaneous Iteration

Before discussing our proof of Theorem 1, we review prior work on Simultaneous Iteration to demonstrate how it can achieve the spectral norm guarantee (2).

Algorithms for Frobenius norm error (1) typically work by *sketching* $\mathbf{A}$ into very few dimensions using a Johnson-Lindenstrauss random projection matrix $\mathbf{\Pi}$ with $\mathrm{poly}(k/\epsilon)$ columns.

$$\mathbf{A}_{n \times d} \times \mathbf{\Pi}_{d \times \mathrm{poly}(k/\epsilon)} = (\mathbf{A}\mathbf{\Pi})_{n \times \mathrm{poly}(k/\epsilon)}$$

$\mathbf{\Pi}$ is usually a random Gaussian or (possibly sparse) random sign matrix and $\mathbf{Z}$ is computed using the SVD of $\mathbf{A}\mathbf{\Pi}$ or of $\mathbf{A}$ projected onto $\mathbf{A}\mathbf{\Pi}$ [3, 5, 30]. This "sketch-and-solve" approach is very efficient – the computation of $\mathbf{A}\mathbf{\Pi}$ is easily parallelized and, regardless, pass-efficient in a single processor setting. Furthermore, once a small compression of $\mathbf{A}$ is obtained, it can be manipulated in fast memory for the final computation of $\mathbf{Z}$.

However, Frobenius norm error seems an inherent limitation of sketch-and-solve methods. The noise from $\mathbf{A}$'s lower $r - k$ singular values corrupts $\mathbf{A}\mathbf{\Pi}$, making it impossible to extract a good partial SVD if the sum of these singular values (equal to $\|\mathbf{A} - \mathbf{A}_k\|_F^2$) is too large.

In order to achieve spectral norm error (2), Simultaneous Iteration must reduce this noise down to the scale of $\sigma_{k+1} = \|\mathbf{A} - \mathbf{A}_k\|_2$. It does this by working with the powered matrix $\mathbf{A}^q$ [31].[3] By the spectral theorem, $\mathbf{A}^q$ has exactly the same *singular vectors* as $\mathbf{A}$, but its *singular values* are equal to those of $\mathbf{A}$ raised to the $q^{\text{th}}$ power. Powering spreads the values apart and accordingly, $\mathbf{A}^q$'s lower singular values are relatively much smaller than its top singular values (see example in Figure 2a).

Specifically, $q = O(\frac{\log d}{\epsilon})$ is sufficient to increase any singular value $\geq (1 + \epsilon)\sigma_{k+1}$ to be significantly (i.e. $\mathrm{poly}(d)$ times) larger than any value $\leq \sigma_{k+1}$. This effectively denoises our problem – if we use a sketching method to find a good $\mathbf{Z}$ for approximating $\mathbf{A}^q$ up to Frobenius norm error, $\mathbf{Z}$ will have to align very well with every singular vector with value $\geq (1 + \epsilon)\sigma_{k+1}$. It thus provides an accurate basis for approximating $\mathbf{A}$ up to small spectral norm error.

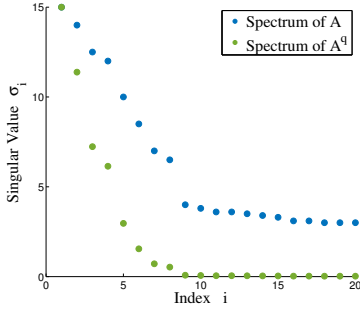

(a) $\mathbf{A}$'s singular values compared to those of $\mathbf{A}^q$, rescaled to match on $\sigma_1$. Notice the significantly reduced tail after $\sigma_8$.

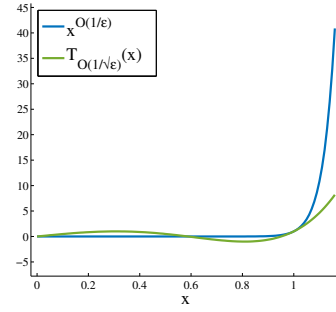

(b) An $O(1/\sqrt{\epsilon})$-degree Chebyshev polynomial, $T_{O(1/\sqrt{\epsilon})}(x)$, pushes low values nearly as close to zero as $x^{O(1/\epsilon)}$.

Figure 2: Replacing $\mathbf{A}$ with a matrix polynomial facilitates higher accuracy approximation.

Computing $\mathbf{A}^q$ directly is costly, so $\mathbf{A}^q\mathbf{\Pi}$ is computed iteratively – start with a random $\mathbf{\Pi}$ and repeatedly multiply by $\mathbf{A}$ on the left. Since even a rough Frobenius norm approximation for $\mathbf{A}^q$ suffices, $\mathbf{\Pi}$ can be chosen to have just $k$ columns. Each iteration thus takes $O(\text{nnz}(\mathbf{A})k)$ time.

When analyzing Simultaneous Iteration, [15] uses the following randomized sketch-and-solve result to find a $\mathbf{Z}$ that gives a coarse Frobenius norm approximation to $\mathbf{B} = \mathbf{A}^q$ and therefore a good spectral norm approximation to $\mathbf{A}$. The lemma is numbered for consistency with our full paper.

**Lemma 4** (Frobenius Norm Low-Rank Approximation). *For any $\mathbf{B} \in \mathbb{R}^{n \times d}$ and $\mathbf{\Pi} \in \mathbb{R}^{d \times k}$ where the entries of $\mathbf{\Pi}$ are independent Gaussians drawn from $\mathcal{N}(0,1)$. If we let $\mathbf{Z}$ be an orthonormal basis for $\text{span}(\mathbf{B}\mathbf{\Pi})$, then with probability at least $99/100$, for some fixed constant $c$,*

$$\|\mathbf{B} - \mathbf{Z}\mathbf{Z}^T\mathbf{B}\|_F^2 \le c \cdot dk\|\mathbf{B} - \mathbf{B}_k\|_F^2.$$

For analyzing block methods, results like Lemma 4 can effectively serve as a replacement for earlier random initialization analysis that applies to single vector power and Krylov methods [32].

$\sigma_{k+1}(\mathbf{A}^q) \le \frac{1}{\text{poly}(d)}\sigma_m(\mathbf{A}^q)$ for any $m$ with $\sigma_m(\mathbf{A}) \ge (1+\epsilon)\sigma_{k+1}(\mathbf{A})$. Plugging into Lemma 4:

$$\|\mathbf{A}^q - \mathbf{Z}\mathbf{Z}^T\mathbf{A}^q\|_F^2 \le cdk \cdot \sum_{i=k+1}^{r} \sigma_i^2(\mathbf{A}^q) \le cdk \cdot d \cdot \sigma_{k+1}^2(\mathbf{A}^q) \le \sigma_m^2(\mathbf{A}^q)/\text{poly}(d).$$

Rearranging using Pythagorean theorem, we have $\|\mathbf{Z}\mathbf{Z}^T\mathbf{A}^q\|_F^2 \ge \|\mathbf{A}^q\|_F^2 - \frac{\sigma_m^2(\mathbf{A}^q)}{\text{poly}(d)}$. That is, $\mathbf{A}^q$'s projection onto $\mathbf{Z}$ captures nearly all of its Frobenius norm. This is only possible if $\mathbf{Z}$ aligns very well with the top singular vectors of $\mathbf{A}^q$ and hence gives a good spectral norm approximation for $\mathbf{A}$.

## 4   Proof Sketch for Theorem 1

The intuition for beating Simultaneous Iteration with Block Krylov Iteration matches that of many accelerated iterative methods. Simply put, there are better polynomials than $\mathbf{A}^q$ for denoising tail singular values. In particular, we can use a *lower degree* polynomial, allowing us to compute fewer powers of $\mathbf{A}$ and thus leading to an algorithm with fewer iterations. For example, an appropriately shifted $q = O(\log(d)/\sqrt{\epsilon})$ degree Chebyshev polynomial can push the tail of $\mathbf{A}$ nearly as close to zero as $\mathbf{A}^{O(\log d/\epsilon)}$, even if the long run growth of the polynomial is much lower (see Figure 2b).

Specifically, we prove the following scalar polynomial lemma in the full version of our paper [25], which can then be applied to effectively denoising $\mathbf{A}$'s singular value tail.

**Lemma 5** (Chebyshev Minimizing Polynomial). *For $\epsilon \in (0,1]$ and $q = O(\log d/\sqrt{\epsilon})$, there exists a degree $q$ polynomial $p(x)$ such that $p((1+\epsilon)\sigma_{k+1}) = (1+\epsilon)\sigma_{k+1}$ and,*

*1)  $p(x) \ge x$ for $x \ge (1+\epsilon)\sigma_{k+1}$     2)  $|p(x)| \le \frac{\sigma_{k+1}}{\text{poly}(d)}$ for $x \le \sigma_{k+1}$.*

*Furthermore, we can choose the polynomial to only contain monomials with odd powers.*

Block Krylov Iteration takes advantage of such polynomials by working with the Krylov subspace,

$$\mathbf{K} = \begin{bmatrix} \mathbf{\Pi} & \mathbf{A\Pi} & \mathbf{A}^2\mathbf{\Pi} & \mathbf{A}^3\mathbf{\Pi} & \dots & \mathbf{A}^q\mathbf{\Pi} \end{bmatrix},$$

from which we can construct $p_q(\mathbf{A})\mathbf{\Pi}$ for any polynomial $p_q(\cdot)$ of degree $q$.[4] Since the polynomial from Lemma 5 must be scaled and shifted based on the value of $\sigma_{k+1}$, we cannot easily compute it directly. Instead, we argue that the very best $k$ rank approximation to $\mathbf{A}$ lying in the span of $\mathbf{K}$ at least matches the approximation achieved by projecting onto the span of $p_q(\mathbf{A})\mathbf{\Pi}$. Finding this best approximation will therefore give a nearly optimal low-rank approximation to $\mathbf{A}$.

Unfortunately, there's a catch. Surprisingly, it is not clear how to efficiently compute the best spectral norm error low-rank approximation to $\mathbf{A}$ lying in a given subspace (e.g. $\mathbf{K}$'s span) [14, 33]. This challenge precludes an analysis of Krylov methods parallel to recent work on Simultaneous Iteration.

Nevertheless, since our analysis shows that projecting to $\mathbf{Z}$ captures nearly all the Frobenius norm of $p_q(\mathbf{A})$, we can show that the best *Frobenius norm* low-rank approximation to $\mathbf{A}$ in the span of $\mathbf{K}$ gives good enough spectral norm approximation. By the following lemma, this optimal Frobenius norm low-rank approximation is given by $\mathbf{Z}\mathbf{Z}^T\mathbf{A}$, where $\mathbf{Z}$ is exactly the output of Algorithm 2.

**Lemma 6** (Lemma 4.1 of [15]). *Given $\mathbf{A} \in \mathbb{R}^{n \times d}$ and $\mathbf{Q} \in \mathbb{R}^{m \times n}$ with orthonormal columns,*

$$\|\mathbf{A} - (\mathbf{Q}\mathbf{Q}^T\mathbf{A})_k\|_F = \|\mathbf{A} - \mathbf{Q}\left(\mathbf{Q}^T\mathbf{A}\right)_k\|_F = \min_{\mathbf{C}|\mathrm{rank}(\mathbf{C})=k}\|\mathbf{A} - \mathbf{Q}\mathbf{C}\|_F.$$

$\mathbf{Q}\left(\mathbf{Q}^T\mathbf{A}\right)_k$ *can be obtained using an SVD of the $m \times m$ matrix $\mathbf{M} = \mathbf{Q}^T(\mathbf{A}\mathbf{A}^T)\mathbf{Q}$. Specifically, letting $\mathbf{M} = \bar{\mathbf{U}}\bar{\mathbf{\Sigma}}^2\bar{\mathbf{U}}^T$ be the SVD of $\mathbf{M}$, and $\mathbf{Z} = \mathbf{Q}\bar{\mathbf{U}}_k$ then $\mathbf{Q}\left(\mathbf{Q}^T\mathbf{A}\right)_k = \mathbf{Z}\mathbf{Z}^T\mathbf{A}$.*

### 4.1 Stronger Per Vector Error Guarantees

Achieving the per vector guarantee of (3) requires a more nuanced understanding of how Simultaneous Iteration and Block Krylov Iteration denoise the spectrum of $\mathbf{A}$. The analysis for spectral norm low-rank approximation relies on the fact that $\mathbf{A}^q$ (or $p_q(\mathbf{A})$ for Block Krylov Iteration) blows up any singular value $\geq (1+\epsilon)\sigma_{k+1}$ to much larger than any singular value $\leq \sigma_{k+1}$. This ensures that our output $\mathbf{Z}$ aligns very well with the singular vectors corresponding to these large singular values.

If $\sigma_k \geq (1+\epsilon)\sigma_{k+1}$, then $\mathbf{Z}$ aligns well with all top $k$ singular vectors of $\mathbf{A}$ and we get good Frobenius norm error and the per vector guarantee (3). Unfortunately, when there is a small gap between $\sigma_k$ and $\sigma_{k+1}$, $\mathbf{Z}$ could miss intermediate singular vectors whose values lie between $\sigma_{k+1}$ and $(1+\epsilon)\sigma_{k+1}$. This is the case where gap dependent guarantees of classical analysis break down.

However, $\mathbf{A}^q$ or, for Block Krylov Iteration, some $q$-degree polynomial in our Krylov subspace, also significantly separates singular values $> \sigma_{k+1}$ from those $< (1-\epsilon)\sigma_{k+1}$. Thus, each column of $\mathbf{Z}$ at least aligns with $\mathbf{A}$ nearly as well as $\mathbf{u}_{k+1}$. So, even if we miss singular values between $\sigma_{k+1}$ and $(1+\epsilon)\sigma_{k+1}$, they will be replaced with approximate singular values $> (1-\epsilon)\sigma_{k+1}$, enough for (3).

For Frobenius norm low-rank approximation (1), we prove that the degree to which $\mathbf{Z}$ falls outside of the span of $\mathbf{A}$'s top $k$ singular vectors depends on the number of singular values between $\sigma_{k+1}$ and $(1-\epsilon)\sigma_{k+1}$. These are the values that could be 'swapped in' for the true top $k$ singular values. Since their weight counts towards $\mathbf{A}$'s tail, our total loss compared to optimal is at worst $\epsilon\|\mathbf{A} - \mathbf{A}_k\|_F^2$.

## 5 Implementation and Runtimes

For both Algorithm 1 and 2, $\mathbf{\Pi}$ can be replaced by a random sign matrix, or any matrix achieving the guarantee of Lemma 4. $\mathbf{\Pi}$ may also be chosen with $p > k$ columns. In our full paper [25], we discuss in detail how this approach can give improved accuracy.

### 5.1 Simultaneous Iteration

In our implementation we set $\mathbf{Z} = \mathbf{Q}\bar{\mathbf{U}}_k$, which is necessary for achieving per vector guarantees for approximate PCA. However, for near optimal low-rank approximation, we can simply set $\mathbf{Z} = \mathbf{Q}$. Projecting $\mathbf{A}$ to $\mathbf{Q}\bar{\mathbf{U}}_k$ is equivalent to projecting to $\mathbf{Q}$ as these matrices have the same column spans.

Since powering $\mathbf{A}$ spreads its singular values, $\mathbf{K} = (\mathbf{A}\mathbf{A}^T)^q\mathbf{A}\mathbf{\Pi}$ could be poorly conditioned. To improve stability we orthonormalize $\mathbf{K}$ after every iteration (or every few iterations). This does not change $\mathbf{K}$'s column span, so it gives an equivalent algorithm in exact arithmetic.

**Theorem 7** (Simultaneous Iteration Runtime). *Algorithm 1 runs in time*

$$O\left(\mathrm{nnz}(\mathbf{A})k\log(d)/\epsilon + nk^2\log(d)/\epsilon\right).$$

*Proof.* Computing $\mathbf{K}$ requires first multiplying $\mathbf{A}$ by $\mathbf{\Pi}$, which takes $O(\mathrm{nnz}(\mathbf{A})k)$ time. Computing $\left(\mathbf{A}\mathbf{A}^T\right)^i \mathbf{A}\mathbf{\Pi}$ given $\left(\mathbf{A}\mathbf{A}^T\right)^{i-1} \mathbf{A}\mathbf{\Pi}$ then takes $O(\mathrm{nnz}(\mathbf{A})k)$ time to first multiply our $(n \times k)$ matrix by $\mathbf{A}^T$ and then by $\mathbf{A}$. Reorthogonalizing after each iteration takes $O(nk^2)$ time via Gram-Schmidt. This gives a total runtime of $O(\mathrm{nnz}(\mathbf{A})kq + nk^2q)$ for computing $\mathbf{K}$. Finding $\mathbf{Q}$ takes $O(nk^2)$ time. Computing $\mathbf{M}$ by multiplying from left to right requires $O(nnz(\mathbf{A})k + nk^2)$ time. $\mathbf{M}$'s SVD then requires $O(k^3)$ time using classical techniques. Finally, multiplying $\bar{\mathbf{U}}_k$ by $\mathbf{Q}$ takes time $O(nk^2)$. Setting $q = \Theta(\log d/\epsilon)$ gives the claimed runtime. $\qquad\square$

## 5.2 Block Krylov Iteration

In the traditional Block Lanczos algorithm, one starts by computing an orthonormal basis for $\mathbf{A}\mathbf{\Pi}$, the first block in $\mathbf{K}$. Bases for subsequent blocks are computed from previous blocks using a three term recurrence that ensures $\mathbf{Q}^T\mathbf{A}\mathbf{A}^T\mathbf{Q}$ is block tridiagonal, with $k \times k$ sized blocks [18]. This technique can be useful if $qk$ is large, since it is faster to compute the top singular vectors of a block tridiagonal matrix. However, computing $\mathbf{Q}$ using a recurrence can introduce a number of stability issues, and additional steps may be required to ensure that the matrix remains orthogonal [28].

An alternative, uesd in [1], [19], and our Algorithm 2, is to compute $\mathbf{K}$ explicitly and then find $\mathbf{Q}$ using a QR decomposition. This method does not guarantee that $\mathbf{Q}^T\mathbf{A}\mathbf{A}^T\mathbf{Q}$ is block tridiagonal, but avoids stability issues. Furthermore, if $qk$ is small, taking the SVD of $\mathbf{Q}^T\mathbf{A}\mathbf{A}^T\mathbf{Q}$ will still be fast and typically dominated by the cost of computing $\mathbf{K}$.

As with Simultaneous Iteration, we orthonormalize each block of $\mathbf{K}$ after it is computed, avoiding poorly conditioned blocks and giving an equivalent algorithm in exact arithmetic.

**Theorem 8** (Block Krylov Iteration Runtime). *Algorithm 2 runs in time*

$$O\left(\mathrm{nnz}(\mathbf{A})k\log(d)/\sqrt{\epsilon} + nk^2\log^2(d)/\epsilon + k^3\log^3(d)/\epsilon^{3/2}\right).$$

*Proof.* Computing $\mathbf{K}$, including reorthogonalization, requires $O(\mathrm{nnz}(\mathbf{A})kq + nk^2q)$ time. The remaining steps are analogous to those in Simultaneous Iteration except somewhat more costly as we work with a $k \cdot q$ rather than $k$ dimensional subspace. Finding $\mathbf{Q}$ takes $O(n(kq)^2)$ time. Computing $\mathbf{M}$ take $O(nnz(\mathbf{A})(kq) + n(kq)^2)$ time and its SVD then requires $O((kq)^3)$ time. Finally, multiplying $\bar{\mathbf{U}}_k$ by $\mathbf{Q}$ takes time $O(nk(kq))$. Setting $q = \Theta(\log d/\sqrt{\epsilon})$ gives the claimed runtime. $\qquad\square$

# 6 Experiments

We close with several experimental results. A variety of empirical papers, not to mention widespread adoption, already justify the use of randomized SVD algorithms. Prior work focuses in particular on benchmarking Simultaneous Iteration [19, 11] and, due to its improved accuracy over sketch-and-solve approaches, this algorithm is popular in practice [10, 16]. As such, we focus on demonstrating that for many data problems Block Krylov Iteration can offer significantly better convergence.

We implement both algorithms in MATLAB using Gaussian random starting matrices with exactly $k$ columns. We explicitly compute $\mathbf{K}$ for both algorithms, as described in Section 5, and use re-orthonormalization at each iteration to improve stability [34]. We test the algorithms with varying iteration count $q$ on three common datasets, SNAP/AMAZON0302 [22, 23], SNAP/EMAIL-ENRON [22, 35], and 20 NEWSGROUPS [36], computing column principal components in all cases. We plot error vs. iteration count for metrics (1), (2), and (3) in Figure 3. For per vector error (3), we plot the maximum deviation amongst all top $k$ approximate principal components (relative to $\sigma_{k+1}$).

Unsurprisingly, both algorithms obtain very accurate Frobenius norm error, $\|\mathbf{A} - \mathbf{Z}\mathbf{Z}^T\mathbf{A}\|_F/\|\mathbf{A} - \mathbf{A}_k\|_F$, with very few iterations. This is our intuitively weakest guarantee and, in the presence of a heavy singular value tail, both iterative algorithms will outperform the worst case analysis.

On the other hand, for spectral norm low-rank approximation and per vector error, we confirm that Block Krylov Iteration converges much more rapidly than Simultaneous Iteration, as predicted by

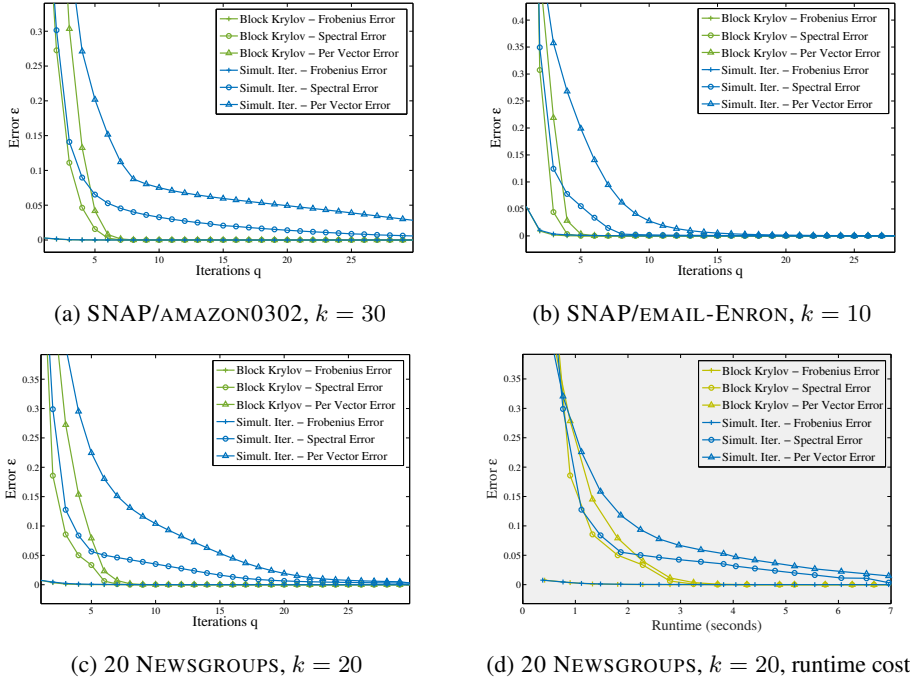

(a) SNAP/AMAZON0302, $k = 30$

(b) SNAP/EMAIL-ENRON, $k = 10$

(c) 20 NEWSGROUPS, $k = 20$

(d) 20 NEWSGROUPS, $k = 20$, runtime cost

Figure 3: Low-rank approximation and per vector error convergence rates for Algorithms 1 and 2.

our theoretical analysis. It it often possible to achieve nearly optimal error with $< 8$ iterations where as getting to within say $1\%$ error with Simultaneous Iteration can take much longer.

The final plot in Figure 3 shows error verses runtime for the $11269 \times 15088$ dimensional 20 NEWS-GROUPS dataset. We averaged over 7 trials and ran the experiments on a commodity laptop with 16GB of memory. As predicted, because its additional memory overhead and post-processing costs are small compared to the cost of the large matrix multiplication required for each iteration, Block Krylov Iteration outperforms Simultaneous Iteration for small $\epsilon$.

More generally, these results justify the importance of convergence bounds that are independent of singular value gaps. Our analysis in Section 6 of the full paper predicts that, once $\epsilon$ is small in comparison to the gap $\frac{\sigma_k}{\sigma_{k+1}} - 1$, we should see much more rapid convergence since $q$ will depend on $\log(1/\epsilon)$ instead of $1/\epsilon$. However, for Simultaneous Iteration, we do not see this behavior with SNAP/AMAZON0302 and it only just begins to emerge for 20 NEWSGROUPS.

While all three datasets have rapid singular value decay, a careful look confirms that their singular value gaps are actually quite small! For example, $\sigma_k/\sigma_{k+1} - 1$ is .004 for SNAP/AMAZON0302 and .011 for 20 NEWSGROUPS, in comparison to .042 for SNAP/EMAIL-ENRON. Accordingly, the frequent claim that singular value gaps can be taken as constant is insufficient, even for small $\epsilon$.

## Footnotes

[1] Here $\mathrm{nnz}(\mathbf{A})$ is the number of non-zero entries in $\mathbf{A}$ and this runtime hides lower order terms.

[2]In fact, it does not even imply $(1 + \epsilon)$ Frobenius norm error.

[3]For nonsymmetric matrices we work with $(\mathbf{A}\mathbf{A}^T)^q\mathbf{A}$, but present the symmetric case here for simplicity.

[4]Algorithm 2 in fact only constructs odd powered terms in $\mathbf{K}$, which is sufficient for our choice of $p_q(x)$.

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
