[Supplementary Material]

# Randomized Block Krylov Methods for Stronger and Faster Approximate Singular Value Decomposition

**Cameron Musco**
Massachusetts Institute of Technology, EECS
Cambridge, MA 02139, USA
cnmusco@mit.edu

**Christopher Musco**
Massachusetts Institute of Technology, EECS
Cambridge, MA 02139, USA
cpmusco@mit.edu

## Abstract

Since being analyzed by Rokhlin, Szlam, and Tygert [1] and popularized by Halko, Martinsson, and Tropp [2], randomized Simultaneous Power Iteration has become the method of choice for approximate singular value decomposition. It is more accurate than simpler sketching algorithms, yet still converges quickly for *any* matrix, independently of singular value gaps. After $\tilde{O}(1/\epsilon)$ iterations, it gives a low-rank approximation within $(1 + \epsilon)$ of optimal for spectral norm error.

We give the first provable runtime improvement on Simultaneous Iteration: a simple randomized block Krylov method, closely related to the classic Block Lanczos algorithm, gives the same guarantees in just $\tilde{O}(1/\sqrt{\epsilon})$ iterations and performs substantially better experimentally. Despite their long history, our analysis is the first of a Krylov subspace method that does not depend on singular value gaps, which are unreliable in practice.

Furthermore, while it is a simple accuracy benchmark, even $(1 + \epsilon)$ error for spectral norm low-rank approximation does not imply that an algorithm returns high quality principal components, a major issue for data applications. We address this problem for the first time by showing that both Block Krylov Iteration and a minor modification of Simultaneous Iteration give nearly optimal PCA for any matrix. This result further justifies their strength over non-iterative sketching methods.

Finally, we give insight beyond the worst case, justifying why both algorithms can run much faster in practice than predicted. We clarify how simple techniques can take advantage of common matrix properties to significantly improve runtime.

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

 a similar phenomenon for the popular 20 NEWSGROUPS dataset [25] and several others. Additionally, the potential failure of low rank approximation measures was recently raised in [22].

We address this issue by introducing a per vector guarantee that requires each approximate singular vector $\mathbf{z}_1, \ldots, \mathbf{z}_k$ to capture nearly as much variance as the corresponding true singular vector:

$$\text{Per Vector Error:} \qquad \forall i, \ \left|\mathbf{u}_i^T \mathbf{A}\mathbf{A}^T \mathbf{u}_i - \mathbf{z}_i^T \mathbf{A}\mathbf{A}^T \mathbf{z}_i\right| \leq \epsilon \sigma_{k+1}^2. \tag{3}$$

The error bound (3) is very strong in that it depends on $\epsilon \sigma_{k+1}^2$, meaning that it is better then relative error, i.e. $\left|\mathbf{u}_i^T \mathbf{A}\mathbf{A}^T \mathbf{u}_i - \mathbf{z}_i^T \mathbf{A}\mathbf{A}^T \mathbf{z}_i\right| \leq \epsilon \sigma_i^2$, for $\mathbf{A}$'s large singular vectors. While it is reminiscent of the bounds sought in classical numerical analysis [26], we stress that it does not require each $\mathbf{z}_i$ to converge to $\mathbf{u}_i$ in the presence of small singular value gaps. In fact, we show that both randomized

Figure 1: Poor per vector error (3) for SNAP/AMAZON0302 returned by a sketch-and-solve approximate SVD that gives very good low-rank approximation in both spectral and Frobenius norm.

Block Krylov Iteration and our slightly modified Simultaneous Iteration algorithm[5] achieve (3) in gap-independent runtimes.

### 2.3 Main Result

Our contributions are summarized in Theorem 1, whose proof appears in parts as Theorems 6 and 7 in Section 5 (runtime) and Theorems 10, 11, and 12 in Section 6 (accuracy).

**Theorem 1** (Main Theorem). *With high probability, Algorithms 1 and 2 find approximate singular vectors $\mathbf{Z} = [\mathbf{z}_1, \ldots, \mathbf{z}_k]$ satisfying guarantees* (1) *and* (2) *for low-rank approximation and* (3) *for PCA. For error $\epsilon$, Algorithm 1 requires $q = O(\log d/\epsilon)$ iterations while Algorithm 2 requires $q = O(\log d/\sqrt{\epsilon})$ iterations. Excluding lower order terms, both algorithms run in time $O(\mathrm{nnz}(\mathbf{A})kq)$.*

We note that, while Simultaneous Iteration was known to achieve (2) [14], surprisingly we are first to prove that it gives (1), a qualitatively weaker goal.

In Section 7 we use our results to give an alternative analysis of both algorithms that *does* depend on singular value gaps and can offer significantly faster convergence when $\mathbf{A}$ has decaying singular values. It is possible to take further advantage of this result by running Algorithms 1 and 2 with a $\mathbf{\Pi}$ that has $> k$ columns, a simple modification for accelerating either method.

Finally, Section 8 contains a number of experiments on large data problems. We justify the importance of gap independent bounds for predicting algorithm convergence and we show that Block Krylov Iteration in fact significantly outperforms the more popular Simultaneous Iteration.

### 2.4 Comparison to Classical Bounds

Decades of work has produced a variety of gap *dependent* bounds for power iteration and Krylov subspace methods. We refer the reader to Saad's standard reference [27]. Most relevant to our work are bounds for block Krylov methods with block size equal to $k$ [28]. Roughly speaking, with randomized initialization, these results offer guarantees equivalent to our strong equation (3) for the top $k$ singular directions after:

$$O\left( \frac{\log(d/\epsilon)}{\sqrt{\frac{\sigma_k}{\sigma_{k+1}} - 1}} \right) \text{ iterations.}$$

This bound is recovered by our Section 7 results and, when the target accuracy $\epsilon$ is smaller than the relative singular value gap $(\sigma_k/\sigma_{k+1} - 1)$, it is tighter than our gap independent results. However, as discussed in Section 8, for high dimensional data problems where $\epsilon$ is set far above machine precision, gap independent bounds more accurately predict required iteration count.

Less comparable to our results are attempts to analyze algorithms with block size *smaller* than $k$ [26]. While "small block" or single vector algorithms offer runtime advantages, it is well understood that with $b$ duplicate singular values, it is impossible to recover the top $k$ singular directions with a block of size $< b$ [29]. More generally, large singular value clusters slow convergence, so any small block algorithm must have runtime dependence on the gaps between *each adjacent pair of top $k$ singular values* [30]. We believe that obtaining simpler theoretical bounds for small block methods is an interesting direction for future work.

# 3   Background and Intuition

We will start by 1) providing background on algorithms for approximate singular value decomposition and 2) giving intuition for Simultaneous Power Iteration and Block Krylov methods and justifying why they can give strong gap-independent error guarantees.

## 3.1   Frobenius Norm Error

Progress on algorithms for Frobenius norm error low-rank approximation (1) has been considerable. Work in this direction dates back to the strong rank-revealing QR factorizations of Gu and Eisenstat [31]. They give deterministic algorithms that run in approximately $O(ndk)$ time, vs. $O(nd^2)$ for a full SVD, but only guarantee polynomial factor Frobenius norm error.

Recently, randomization has been applied to achieve even faster algorithms with $(1 + \epsilon)$ error. The paradigm is to compute a *linear sketch* of $\mathbf{A}$ into very few dimensions using either a column sampling matrix or Johnson-Lindenstrauss random projection matrix $\mathbf{\Pi}$. Typically $\mathbf{A\Pi}$ has at most $\mathrm{poly}(k/\epsilon)$ columns and can be used to quickly find $\mathbf{Z}$. Specifically, $\mathbf{Z}$ is typically taken to be the top $k$ left singular vectors of $\mathbf{A\Pi}$ or of $\mathbf{A}$ projected onto $\mathbf{A\Pi}$ [32, 4].

$$\mathbf{A}_{n \times d} \times \mathbf{\Pi}_{d \times \mathrm{poly}(k/\epsilon)} = (\mathbf{A\Pi})_{n \times \mathrm{poly}(k/\epsilon)}$$

This approach was developed and refined in several pioneering results, including [33, 34, 35, 36] for column sampling, [37, 5] for random projection, and definitive work by Sarlós [4]. Recent work on sparse Johnson-Lindenstrauss type matrices [6, 38, 39] has significantly reduced the cost of multiplying $\mathbf{A\Pi}$, bringing the cost of Frobenius error low-rank approximation down to $O(\mathrm{nnz}(\mathbf{A}) + n\,\mathrm{poly}(k/\epsilon))$ time, where the first term is considered to dominate since typically $k \ll n, d$.

The sketch-and-solve method is very efficient – the computation of $\mathbf{A\Pi}$ is easily parallelized and, regardless, pass-efficient in a single processor setting. Furthermore, once a small compression of $\mathbf{A}$ is obtained, it can be manipulated in fast memory to find $\mathbf{Z}$. This is not typically true of $\mathbf{A}$ itself, making it difficult to directly process the original matrix at all.

## 3.2   Spectral Norm Error via Simultaneous Iteration

Unfortunately, as discussed, Frobenius norm error is often insufficient when $\mathbf{A}$ has a heavy singular value tail. Moreover, it seems an inherent limitation of sketch-and-solve methods. The noise from $\mathbf{A}$'s lower $r - k$ singular values corrupts $\mathbf{A\Pi}$, making it impossible to extract a good partial SVD if the sum of these singular values (equal to $\|\mathbf{A} - \mathbf{A}_k\|_F^2$) is too large. In other words, any error inherently depends on the size of this tail.

In order to achieve spectral norm error (2), Simultaneous Iteration must reduce this noise down to the scale of $\sigma_{k+1} = \|\mathbf{A} - \mathbf{A}_k\|_2$. It does this by working with the powered matrix $\mathbf{A}^q$ [40, 41].[6] By the spectral theorem, $\mathbf{A}^q$ has exactly the same *singular vectors* as $\mathbf{A}$, but its *singular values* are equal to the singular values of $\mathbf{A}$ raised to the $q^{\text{th}}$ power. Powering spreads the values apart and accordingly, $\mathbf{A}^q$'s lower singular values are relatively much smaller than its top singular values (see Figure 2a for an example).

Specifically, $q = O(\frac{\log d}{\epsilon})$ is sufficient to increase any singular value $\geq (1 + \epsilon)\sigma_{k+1}$ to be significantly (i.e. $\mathrm{poly}(d)$ times) larger than any value $\leq \sigma_{k+1}$. This effectively denoises our problem – if we use a sketching method to find a good $\mathbf{Z}$ for approximating $\mathbf{A}^q$ up to Frobenius norm error, $\mathbf{Z}$ will have to align very well with every singular vector with value $\geq (1 + \epsilon)\sigma_{k+1}$. It thus provides an accurate basis for approximating $\mathbf{A}$ up to small spectral norm error.

(a) $\mathbf{A}$'s singular values compared to those of $\mathbf{A}^q$, rescaled to match on $\sigma_1$. Notice the significantly reduced tail after $\sigma_8$.

(b) An $O(1/\sqrt{\epsilon})$-degree Chebyshev polynomial, $T_{O(1/\sqrt{\epsilon})}(x)$, pushes low values nearly as close to zero as $x^{O(1/\epsilon)}$ while spreading higher values less significantly.

Figure 2: Replacing $\mathbf{A}$ with a matrix polynomial facilitates higher accuracy approximation.

Computing $\mathbf{A}^q$ directly is costly, so $\mathbf{A}^q\mathbf{\Pi}$ is computed iteratively. We start with a random $\mathbf{\Pi}$ and repeatedly multiply by $\mathbf{A}$ on the left. Since even a rough Frobenius norm approximation for $\mathbf{A}^q$ suffices, $\mathbf{\Pi}$ is often chosen to have just $k$ columns. Each iteration thus takes $O(\text{nnz}(\mathbf{A})k)$ time. After $\mathbf{A}^q\mathbf{\Pi}$ is computed, $\mathbf{Z}$ can simply be set to a basis for its column span.

To the best of our knowledge, this approach to analyzing Simultaneous Iteration without dependence on singular value gaps began with [1]. The technique was popularized in [2] and its analysis improved in [15] and [16]. [14] gives the first bound that directly achieves (2) with $O(\log d/\epsilon)$ power iterations. All of these papers rely on an improved understanding of the benefits of starting with a randomized $\mathbf{\Pi}$, which has developed from work on the sketch-and-solve paradigm.

### 3.3 Beating Simultaneous Iteration with Krylov Methods

As mentioned, numerous papers hint at the possibility of beating Simultaneous Iteration with block Krylov methods [18, 19, 28]. In particular, [1], [20] and [21] suggest and experimentally confirm the potential of a randomized variant of the Block Lanczos algorithm, which we refer to as Block Krylov Iteration (Algorithm 2). However, none of these papers give theoretical bounds on the algorithm's performance.

The intuition behind Block Krylov Iteration matches that of many accelerated iterative methods. Simply put, there are better polynomials than $\mathbf{A}^q$ for denoising tail singular values. In particular, we can use a *lower degree* polynomial, allowing us to compute fewer powers of $\mathbf{A}$ and thus leading to an algorithm with fewer iterations. For example, an appropriately shifted $q = O(\frac{\log d}{\sqrt{\epsilon}})$ degree Chebyshev polynomial can push the tail of $\mathbf{A}$ nearly as close to zero as $\mathbf{A}^{O(\log d/\epsilon)}$, even if the long run growth of the polynomial is much lower (see Figure 2b).

Block Krylov Iteration takes advantage of such polynomials by working with the Krylov subspace,

$$\mathbf{K} = \begin{bmatrix} \mathbf{\Pi} & \mathbf{A\Pi} & \mathbf{A}^2\mathbf{\Pi} & \mathbf{A}^3\mathbf{\Pi} & \dots & \mathbf{A}^q\mathbf{\Pi} \end{bmatrix},$$

from which we can construct $p_q(\mathbf{A})\mathbf{\Pi}$ for any polynomial $p_q(\cdot)$ of degree $q$.[7] Since an effective polynomial for denoising $\mathbf{A}$ must be scaled and shifted based on the value of $\sigma_{k+1}$, we cannot easily compute it directly. Instead, we argue that the very best $k$ rank approximation to $\mathbf{A}$ lying in the span of $\mathbf{K}$ at least matches the approximation achieved by projecting onto the span of $p_q(\mathbf{A})\mathbf{\Pi}$. Finding this best approximation will therefore give a nearly optimal low-rank approximation to $\mathbf{A}$.

Unfortunately, there's a catch. Perhaps surprisingly, it is not clear how to efficiently compute the best spectral norm error low-rank approximation to $\mathbf{A}$ lying in a specific subspace (e.g. $\mathbf{K}$'s span) [16, 42]. This challenge precludes an analysis of Krylov methods parallel to the recent work on

Simultaneous Iteration. Nevertheless, we show that computing the best Frobenius error low-rank approximation in the span of $\mathbf{K}$, exactly the post-processing step taken by classic Block Lanczos and our method, will give a good enough spectral norm approximation for achieving $(1 + \epsilon)$ error.

### 3.4 Stronger Per Vector Error Guarantees

Achieving the per vector guarantee of (3) requires a more nuanced understanding of how Simultaneous Iteration and Block Krylov Iteration denoise the spectrum of $\mathbf{A}$. The analysis for spectral norm low-rank approximation relies on the fact that $\mathbf{A}^q$ (or $p_q(\mathbf{A})$ for Block Krylov Iteration) blows up any singular value $\geq (1 + \epsilon)\sigma_{k+1}$ to much larger than any singular value $\leq \sigma_{k+1}$. This ensures that the $\mathbf{Z}$ outputted by both algorithms aligns very well with the singular vectors corresponding to these large singular values.

If $\sigma_k \geq (1 + \epsilon)\sigma_{k+1}$, then $\mathbf{Z}$ aligns well with all top $k$ singular vectors of $\mathbf{A}$ and we get good Frobenius norm error and the per vector guarantee (3). Unfortunately, when there is a small gap between $\sigma_k$ and $\sigma_{k+1}$, $\mathbf{Z}$ could miss intermediate singular vectors whose values lie between $\sigma_{k+1}$ and $(1 + \epsilon)\sigma_{k+1}$. This is the case where gap dependent guarantees of classical analysis break down.

However, $\mathbf{A}^q$ or, for Block Krylov Iteration, some $q$-degree polynomial in our Krylov subspace, also significantly separates singular values $> \sigma_{k+1}$ from those $< (1 - \epsilon)\sigma_{k+1}$. Thus, each column of $\mathbf{Z}$ at least aligns with $\mathbf{A}$ nearly as well as $\mathbf{u}_{k+1}$. So, even if we miss singular values between $\sigma_{k+1}$ and $(1 + \epsilon)\sigma_{k+1}$, they will be replaced with approximate singular values $> (1 - \epsilon)\sigma_{k+1}$, enough for (3).

For Frobenius norm low-rank approximation, we prove that the degree to which $\mathbf{Z}$ falls outside of the span of $\mathbf{A}$'s top $k$ singular vectors depends on the number of singular values between $\sigma_{k+1}$ and $(1 - \epsilon)\sigma_{k+1}$. These are the values that could be 'swapped in' for the true top $k$ singular values. Since their weight counts towards $\mathbf{A}$'s tail, our total loss compared to optimal is at worst $\epsilon \|\mathbf{A} - \mathbf{A}_k\|_F^2$.

## 4 Preliminaries

Before proceeding to the full technical analysis, we overview required results from linear algebra, polynomial approximation, and randomized low-rank approximation.

### 4.1 Singular Value Decomposition and Low-Rank Approximation

Using the SVD, we compute the pseudoinverse of $\mathbf{A} \in \mathbb{R}^{n \times d}$ as $\mathbf{A}^+ = \mathbf{V}\boldsymbol{\Sigma}^{-1}\mathbf{U}^T$. Additionally, for any polynomial $p(x)$, we define $p(\mathbf{A}) = \mathbf{U}p(\boldsymbol{\Sigma})\mathbf{V}^{\mathbf{T}}$. Note that, since singular values are always take to be non-negative, $p(\mathbf{A})$'s singular values are given by $|p(\boldsymbol{\Sigma})|$.

Let $\boldsymbol{\Sigma}_k$ be $\boldsymbol{\Sigma}$ with all but its largest $k$ singular values zeroed out. Let $\mathbf{U}_k$ and $\mathbf{V}_k$ be $\mathbf{U}$ and $\mathbf{V}$ with all but their first $k$ columns zeroed out. For any $k$, $\mathbf{A}_k = \mathbf{U}\boldsymbol{\Sigma}_k\mathbf{V}^{\mathbf{T}} = \mathbf{U}_k\boldsymbol{\Sigma}_k\mathbf{V}_k^T$ is the closest rank $k$ approximation to $\mathbf{A}$ for any unitarily invariant norm, including the Frobenius norm and spectral norm [43]. The squared Frobenius norm is given by $\|\mathbf{A}\|_F^2 = \sum_{i,j} \mathbf{A}_{i,j}^2 = \mathrm{tr}(\mathbf{A}\mathbf{A}^{\mathbf{T}}) = \sum_i \sigma_i^2$. The spectral norm is given by $\|\mathbf{A}\|_2 = \sigma_1$.

$$\|\mathbf{A} - \mathbf{A}_k\|_F = \min_{\mathbf{B}|\mathrm{rank}(\mathbf{B})=k} \|\mathbf{A} - \mathbf{B}\|_F \quad \text{and} \quad \|\mathbf{A} - \mathbf{A}_k\|_2 = \min_{\mathbf{B}|\mathrm{rank}(\mathbf{B})=k} \|\mathbf{A} - \mathbf{B}\|_2.$$

We often work with the remainder matrix $\mathbf{A} - \mathbf{A}_k$ and label it $\mathbf{A}_{r \setminus k}$. Its singular value decomposition is given by $\mathbf{A}_{r \setminus k} = \mathbf{U}_{r \setminus k}\boldsymbol{\Sigma}_{r \setminus k}\mathbf{V}_{r \setminus k}^T$ where $\mathbf{U}_{r \setminus k}$, $\boldsymbol{\Sigma}_{r \setminus k}$, and $\mathbf{V}_{r \setminus k}^T$ have their first $k$ columns zeroed.

While the SVD gives a globally optimal rank $k$ approximation for $\mathbf{A}$, both Simultaneous Iteration and Block Krylov Iteration return the best $k$ rank approximation falling within some fixed subspace spanned by a basis $\mathbf{Q}$ (with rank $\geq k$). For the Frobenius norm, this simply requires projecting $\mathbf{A}$ to $\mathbf{Q}$ and taking the best rank $k$ approximation of the resulting matrix using an SVD.

**Lemma 2** (Lemma 4.1 of [14]). *Given $\mathbf{A} \in \mathbb{R}^{n \times d}$ and $\mathbf{Q} \in \mathbb{R}^{m \times n}$ with orthonormal columns,*

$$\|\mathbf{A} - (\mathbf{Q}\mathbf{Q}^T\mathbf{A})_k\|_F = \|\mathbf{A} - \mathbf{Q}(\mathbf{Q}^T\mathbf{A})_k\|_F = \min_{\mathbf{C}|\mathrm{rank}(\mathbf{C})=k} \|\mathbf{A} - \mathbf{Q}\mathbf{C}\|_F.$$

*This low-rank approximation can be obtained using an SVD (equivalently, eigendecomposition) of the $m \times m$ matrix $\mathbf{M} = \mathbf{Q}^T(\mathbf{A}\mathbf{A}^T)\mathbf{Q}$. Specifically, letting $\mathbf{M} = \bar{\mathbf{U}}\bar{\boldsymbol{\Sigma}}^2\bar{\mathbf{U}}^T$, then:*

$$\left(\mathbf{Q}\bar{\mathbf{U}}_k\right)\left(\mathbf{Q}\bar{\mathbf{U}}_k\right)^T \mathbf{A} = \mathbf{Q}\left(\mathbf{Q}^T\mathbf{A}\right)_k.$$

If the SVD of $\mathbf{Q}^T\mathbf{A}$ is given by $\mathbf{Q}^T\mathbf{A} = \bar{\mathbf{U}}\bar{\mathbf{\Sigma}}\bar{\mathbf{V}}^T$ then $\mathbf{M} = \mathbf{Q}^T(\mathbf{A}\mathbf{A}^T)\mathbf{Q} = \bar{\mathbf{U}}\bar{\mathbf{\Sigma}}^2\bar{\mathbf{U}}^T$. So $\mathbf{Q}\left(\mathbf{Q}^T\mathbf{A}\right)_k = \mathbf{Q}\bar{\mathbf{U}}_k\bar{\mathbf{\Sigma}}_k\bar{\mathbf{V}}_k^T = \mathbf{Q}\left(\bar{\mathbf{U}}_k\bar{\mathbf{U}}_k^T\right)\bar{\mathbf{U}}\bar{\mathbf{\Sigma}}\bar{\mathbf{V}}^T = \mathbf{Q}\bar{\mathbf{U}}_k\bar{\mathbf{U}}_k^T\mathbf{Q}^T\mathbf{A}$, giving the lower matrix equality. Note that $\mathbf{Q}\bar{\mathbf{U}}_k$ has orthonormal columns since $\bar{\mathbf{U}}_k^T\mathbf{Q}^T\mathbf{Q}\bar{\mathbf{U}}_k = \bar{\mathbf{U}}_k^T\mathbf{I}\bar{\mathbf{U}}_k = \mathbf{I}_k$.

In general, this rank $k$ approximation *does not* give the best spectral norm approximation to $\mathbf{A}$ falling within $\mathbf{Q}$ [16]. A closed form solution can be obtained using the results of [42], which are related to Parrott's theorem, but we do not know how to compute this solution without essentially performing an SVD of $\mathbf{A}$. It is at least simple to show that the optimal spectral norm approximation for $\mathbf{A}$ spanned by a rank $k$ basis is obtained by projecting $\mathbf{A}$ to the basis:

**Lemma 3** (Lemma 4.14 of [14]). *For $\mathbf{A} \in \mathbb{R}^{n \times d}$ and $\mathbf{Q} \in \mathbb{R}^{n \times k}$ with orthonormal columns,*

$$\|\mathbf{A} - \mathbf{Q}\mathbf{Q}^T\mathbf{A}\|_2 = \min_{\mathbf{C}} \|\mathbf{A} - \mathbf{Q}\mathbf{C}\|_2.$$

### 4.2 Other Linear Algebra Tools

Throughout this paper we use $span(\mathbf{M})$ to denote the column span of the matrix $\mathbf{M}$. We say that a matrix $\mathbf{Q}$ is an orthonormal basis for the column span of $\mathbf{M}$ if $\mathbf{Q}$ has orthonormal columns and $\mathbf{Q}\mathbf{Q}^T\mathbf{M} = \mathbf{M}$. That is, projecting the columns of $\mathbf{M}$ to $\mathbf{Q}$ fully recovers those columns. $\mathbf{Q}\mathbf{Q}^T$ is the orthogonal projection matrix onto the span of $\mathbf{Q}$. $(\mathbf{Q}\mathbf{Q}^T)(\mathbf{Q}\mathbf{Q}^T) = \mathbf{Q}\mathbf{I}\mathbf{Q}^T = \mathbf{Q}\mathbf{Q}^T$.

If $\mathbf{M}$ and $\mathbf{N}$ have the same dimension and $\mathbf{M}\mathbf{N}^{\mathbf{T}} = \mathbf{0}$ then $\|\mathbf{M} + \mathbf{N}\|_F^2 = \|\mathbf{M}\|_F^2 + \|\mathbf{N}\|_F^2$. This matrix Pythagorean theorem follows from writing $\|\mathbf{M} + \mathbf{N}\|_F^2 = \mathrm{tr}((\mathbf{M} + \mathbf{N})(\mathbf{M} + \mathbf{N})^{\mathbf{T}})$. As an example, for any orthogonal projection $\mathbf{Q}\mathbf{Q}^T\mathbf{A}$, $\mathbf{A}^T(\mathbf{I} - \mathbf{Q}\mathbf{Q}^T)\mathbf{Q}\mathbf{Q}^T\mathbf{A} = \mathbf{0}$, so $\|\mathbf{A} - \mathbf{Q}\mathbf{Q}^T\mathbf{A}\|_F^2 = \|\mathbf{A}\|_F^2 - \|\mathbf{Q}\mathbf{Q}^T\mathbf{A}\|_F^2$. This implies that, since $\mathbf{A}_k = \mathbf{U}_k\mathbf{U}_k^T\mathbf{A}$ minimizes $\|\mathbf{A} - \mathbf{A}_k\|_F^2$ over all rank $k$ matrices, $\mathbf{Q}\mathbf{Q}^T = \mathbf{U}_k\mathbf{U}_k$ maximizes $\|\mathbf{Q}\mathbf{Q}^T\mathbf{A}\|_F^2$ over all rank $k$ orthogonal projections.

### 4.3 Randomized Low-Rank Approximation

Our proofs build on well known sketch-based algorithms for low-rank approximation with Frobenius norm error. A short proof of the following Lemma is in Appendix A:

**Lemma 4** (Frobenius Norm Low-Rank Approximation). *Take any $\mathbf{A} \in \mathbb{R}^{n \times d}$ and $\mathbf{\Pi} \in \mathbb{R}^{d \times k}$ where the entries of $\mathbf{\Pi}$ are independent Gaussians drawn from $\mathcal{N}(0, 1)$. If we let $\mathbf{Z}$ be an orthonormal basis for $span(\mathbf{A}\mathbf{\Pi})$, then with probability at least $99/100$, for some fixed constant c,*

$$\|\mathbf{A} - \mathbf{Z}\mathbf{Z}^T\mathbf{A}\|_F^2 \leq c \cdot dk\|\mathbf{A} - \mathbf{A}_k\|_F^2.$$

For analyzing block methods, results like Lemma 4 can effectively serve as a replacement for earlier random initialization analysis that applies to single vector power and Krylov methods [44].

### 4.4 Chebyshev Polynomials

As outlined in Section 3.3, our proof also requires polynomials to more effectively denoise the tail of $\mathbf{A}$. As is standard for Krylov subspace methods, we use a variation on the Chebyshev polynomials. The proof of the following Lemma is relegated to Appendix A.

**Lemma 5** (Chebyshev Minimizing Polynomial). *Given a specified value $\alpha > 0$, gap $\gamma \in (0, 1]$, and $q \geq 1$, there exists a degree $q$ polynomial $p(x)$ such that:*

1. $p((1 + \gamma)\alpha) = (1 + \gamma)\alpha$

2. $p(x) \geq x$ for all $x \geq (1 + \gamma)\alpha$

3. $|p(x)| \leq \frac{\alpha}{2^{q\sqrt{\gamma}-1}}$ for all $x \in [0, \alpha]$

*Furthermore, when q is odd, the polynomial only contains odd powered monomials.*

## 5 Implementation and Runtimes

We first briefly discuss runtime and implementation considerations for Algorithms 1 and 2, our randomized implementations of Simultaneous Power Iteration and Block Krylov Iteration.

## 5.1 Simultaneous Iteration

Algorithm 1 can be modified in a number of ways. $\mathbf{\Pi}$ can be replaced by a random sign matrix, or any matrix achieving the guarantee of Lemma 4. $\mathbf{\Pi}$ may also be chosen with $p > k$ columns. We will discuss in detail how this approach can give improved accuracy in Section 7.

In our implementation we set $\mathbf{Z} = \mathbf{Q}\bar{\mathbf{U}}_{\mathbf{k}}$. This ensures that, for all $l \leq k$, $\mathbf{Z}_l$ gives the best rank $l$ Frobenius norm approximation to $\mathbf{A}$ within the span of $\mathbf{K}$ (See Lemma 2). This is necessary for achieving per vector guarantees for approximate PCA. However, if we are only interested in computing a near optimal low-rank approximation, we can simply set $\mathbf{Z} = \mathbf{Q}$. Projecting $\mathbf{A}$ to $\mathbf{Q}\bar{\mathbf{U}}_{\mathbf{k}}$ is equivalent to projecting to $\mathbf{Q}$ as these two matrices have the same column spans.

Additionally, since powering $\mathbf{A}$ spreads its singular values, $\mathbf{K} = (\mathbf{A}\mathbf{A}^T)^q \mathbf{A}\mathbf{\Pi}$ could be poorly conditioned. As suggested in [45], to improve stability we can orthonormalize $\mathbf{K}$ after every iteration (or every few iterations). This does not change $\mathbf{K}$'s column span, so it gives an equivalent algorithm in exact arithmetic, but improves conditioning significantly.

**Theorem 6** (Simultaneous Iteration Runtime). *Algorithm 1 runs in time*

$$O\left(\text{nnz}(\mathbf{A})\frac{k\log d}{\epsilon} + \frac{nk^2 \log d}{\epsilon}\right).$$

*Proof.* Computing $\mathbf{K}$ requires first multiplying $\mathbf{A}$ by $\mathbf{\Pi}$, which takes $O(\text{nnz}(\mathbf{A})k)$ time. Computing $(\mathbf{A}\mathbf{A}^T)^i \mathbf{A}\mathbf{\Pi}$ given $(\mathbf{A}\mathbf{A}^T)^{i-1} \mathbf{A}\mathbf{\Pi}$ then takes $O(\text{nnz}(\mathbf{A})k)$ time to first multiply our $(n \times k)$ matrix by $\mathbf{A}^T$ and then by $\mathbf{A}$. Reorthogonalizing after each iteration takes $O(nk^2)$ time via Gram-Schmidt or Householder reflections. This gives a total runtime of $O(\text{nnz}(\mathbf{A})kq + nk^2q)$ for computing $\mathbf{K}$.

Finding $\mathbf{Q}$ takes $O(nk^2)$ time. Computing $\mathbf{M}$ by multiplying from left to right requires $O(nnz(\mathbf{A})k + nk^2)$ time. $\mathbf{M}$'s SVD then requires $O(k^3)$ time using classical techniques. Finally, multiplying $\bar{\mathbf{U}}_k$ by $\mathbf{Q}$ takes time $O(nk^2)$. Setting $q = \Theta(\log d/\epsilon)$ gives the claimed runtime. $\square$

## 5.2 Block Krylov Iteration

As with Simultaneous Iteration, we can replace $\mathbf{\Pi}$ with any matrix achieving the guarantee of Lemma 4 and can use $p > k$ columns to improve accuracy. $\mathbf{Q}$ can also be computed in a number of ways. In the traditional Block Lanczos algorithm, one starts by computing an orthonormal basis for $\mathbf{A}\mathbf{\Pi}$, the first block in the Krylov subspace. Bases for subsequent blocks are computed from previous blocks using a three term recurrence that ensures $\mathbf{Q}^T\mathbf{A}\mathbf{A}^T\mathbf{Q}$ is block tridiagonal, with $k \times k$ sized blocks [19]. This technique can be useful if $qk$ is large, since it is faster to compute the top singular vectors of a block tridiagonal matrix. However, computing $\mathbf{Q}$ using a recurrence can introduce a number of stability issues, and additional steps may be required to ensure that the matrix remains orthogonal [29].

An alternative is to compute $\mathbf{K}$ explicitly and then compute $\mathbf{Q}$ using a QR decomposition. This method is used in [1] and [20]. It does not guarantee that $\mathbf{Q}^T\mathbf{A}\mathbf{A}^T\mathbf{Q}$ is block tridiagonal, but helps avoid a number of stability issues. Furthermore, if $qk$ is small, taking the SVD of $\mathbf{Q}^T\mathbf{A}\mathbf{A}^T\mathbf{Q}$ will still be fast and typically dominated by the cost of computing $\mathbf{K}$.

As with Simultaneous Iteration, we can also orthonormalize each block of $\mathbf{K}$ after it is computed, avoiding poorly conditioned blocks and giving an equivalent algorithm in exact arithmetic.

**Theorem 7** (Block Krylov Iteration Runtime). *Algorithm 2 runs in time*

$$O\left(\text{nnz}(\mathbf{A})\frac{k\log d}{\sqrt{\epsilon}} + \frac{nk^2 \log^2 d}{\epsilon} + \frac{k^3 \log^3 d}{\epsilon^{3/2}}\right).$$

*Proof.* Computing $\mathbf{K}$, including block reorthogonalization, requires $O(\text{nnz}(\mathbf{A})kq + nk^2q)$ time. The remaining steps are analogous to those in Simultaneous Iteration except somewhat more costly as we work an $k \cdot q$ dimensional rather than $k$ dimensional subspace. Finding $\mathbf{Q}$ takes $O(n(kq)^2)$ time. Computing $\mathbf{M}$ take $O(nnz(\mathbf{A})(kq) + n(kq)^2)$ time and its SVD then requires $O((kq)^3)$ time.

Finally, multiplying $\bar{\mathbf{U}}_k$ by $\mathbf{Q}$ takes time $O(nk(kq))$. Setting $q = \Theta(\log d/\sqrt{\epsilon})$ gives the claimed runtime. $\qquad\square$

## 6 Error Bounds

We next prove that both Algorithms 1 and 2 return a basis $\mathbf{Z}$ that gives relative error Frobenius (1) and spectral norm (2) low-rank approximation error as well as the per vector guarantees (3).

### 6.1 Main Approximation Lemma

We start with a general approximation lemma, which gives three guarantees formalizing the intuition given in Section 3. All other proofs follow nearly immediately from this lemma.

For simplicity we assume that $k \leq r = \text{rank}(\mathbf{A}) \leq n, d$. However, if $k > r$ it can be seen that both algorithms still return a basis satisfying the proven guarantees. We start with a definition:

**Definition 8.** *For a given matrix $\mathbf{Z} \in \mathbb{R}^{n \times k}$ with orthonormal columns, letting $\mathbf{Z}_l \in \mathbb{R}^{n \times l}$ be the first $l$ columns of $\mathbf{Z}$, we define the error function:*

$$\begin{aligned}
\mathcal{E}(\mathbf{Z}_l, \mathbf{A}) &= \|\mathbf{A}_l\|_F^2 - \|\mathbf{Z}_l \mathbf{Z}_l^T \mathbf{A}\|_F^2 \\
&= \|\mathbf{A} - \mathbf{Z}_l \mathbf{Z}_l^T \mathbf{A}\|_F^2 - \|\mathbf{A} - \mathbf{A}_l\|_F^2.
\end{aligned}$$

Recall that $\mathbf{A}_l$ is the best rank $l$ approximation to $\mathbf{A}$. This error function measures how well $\mathbf{Z}_l \mathbf{Z}_l^T \mathbf{A}$ approximates $\mathbf{A}$ in comparison to the optimal.

**Lemma 9** (Main Approximation Lemma). *Let $m$ be the number of singular values $\sigma_i$ of $\mathbf{A}$ with $\sigma_i \geq (1 + \epsilon/2)\sigma_{k+1}$. Let $w$ be the number of singular values with $\frac{1}{1+\epsilon/2}\sigma_k \leq \sigma_i < \sigma_k$. With probability $99/100$ Algorithms 1 and 2 return $\mathbf{Z}$ satisfying:*

1. *$\forall l \leq m, \mathcal{E}(\mathbf{Z}_l, \mathbf{A}) \leq (\epsilon/2) \cdot \sigma_{k+1}^2$,*

2. *$\forall l \leq k, \mathcal{E}(\mathbf{Z}_l, \mathbf{A}) \leq \mathcal{E}(\mathbf{Z}_{l-1}, \mathbf{A}) + 3\epsilon \cdot \sigma_{k+1}^2$,*

3. *$\forall l \leq k, \mathcal{E}(\mathbf{Z}_l, \mathbf{A}) \leq (w + 1) \cdot 3\epsilon \cdot \sigma_{k+1}^2$.*

Property 1 captures the intuition given in Section 3.2. Both algorithms return $\mathbf{Z}$ with $\mathbf{Z}_l$ equal to the best Frobenius norm low-rank approximation in $span(\mathbf{K})$. Since $\sigma_1 \geq \ldots \geq \sigma_m \geq (1 + \epsilon/2)\sigma_{k+1}$ and our polynomials separate any values above this threshold from anything below $\sigma_{k+1}$, $\mathbf{Z}$ must align very well with $\mathbf{A}$'s top $m$ singular vectors. Thus $\mathcal{E}(\mathbf{Z}_l, \mathbf{A})$ is very small for all $l \leq m$.

Property 2 captures the intuition of Section 3.4 – outside of the largest $m$ singular values, $\mathbf{Z}$ still performs well. We may fail to distinguish between vectors with values between $\frac{1}{1+\epsilon/2}\sigma_k$ and $(1 + \epsilon/2)\sigma_{k+1}$. However, aligning with the smaller vectors in this range rather than the larger vectors can incur a cost of at most $O(\epsilon)\sigma_{k+1}^2$. Since every column of $\mathbf{Z}$ outside of the first $m$ may incur such a cost, there is a linear accumulation as characterized by Property 2.

Finally, Property 3 captures the intuition that the total error in $\mathbf{Z}$ is bounded by the number of singular values falling in the range $\frac{1}{1+\epsilon/2}\sigma_k \leq \sigma_i < \sigma_k$. This is the total number of singular vectors that aren't necessarily separated from and can thus be 'swapped in' for any of the $(k - m)$ true top vectors with singular value $< (1 + \epsilon/2)\sigma_{k+1}$. Property 3 is critical in achieving near optimal Frobenius norm low-rank approximation.

*Proof.* **Proof of Property 1**

Assume $m \geq 1$. If $m = 0$ then Property 1 trivially holds. We will prove the statement for Algorithm 2, since this is the more complex case, and then explain how the proof extends to Algorithm 1.

Let $p_1$ be the polynomial from Lemma 5 with $\alpha = \sigma_{k+1}$, $\gamma = \epsilon/2$, and $q \geq c \log(d/\epsilon)/\sqrt{\epsilon}$ for some fixed constant $c$. We can assume $1/\epsilon = O(\text{poly } d)$ and thus $q = O(\log d/\sqrt{\epsilon})$. Otherwise our Krylov subspace would have as many columns as $\mathbf{A}$ and we may as well use a classical algorithm to compute $\mathbf{A}$'s partial SVD directly. Let $\mathbf{Y}_1 \in \mathbb{R}^{n \times k}$ be an orthonormal basis for the span of

$p_1(\mathbf{A})\mathbf{\Pi}$. Recall that we defined $p_1(\mathbf{A}) = \mathbf{U}p_1(\mathbf{\Sigma})\mathbf{V}^T$. As long as we choose $q$ to be odd, by the recursive definition of the Chebyshev polynomials, $p_1(\mathbf{A})$ only contains odd powers of $\mathbf{A}$ (see Lemma 5). Any odd power $i$ can be evaluated as $\left(\mathbf{A}\mathbf{A}^T\right)^{(i-1)/2}\mathbf{A}$. Accordingly, $p_1(\mathbf{A})\mathbf{\Pi}$ and thus $\mathbf{Y}_1$ have columns falling within the span of the Krylov subspace from Algorithm 2 (and hence its column basis $\mathbf{Q}$).

By Lemma 4 we have with probability $99/100$:

$$\|p_1(\mathbf{A}) - \mathbf{Y}_1\mathbf{Y}_1^T p_1(\mathbf{A})\|_F^2 \leq cdk\|p_1(\mathbf{A}) - p_1(\mathbf{A})_k\|_F^2. \tag{4}$$

Furthermore, one possible rank $k$ approximation of $p_1(\mathbf{A})$ is $p_1(\mathbf{A}_k)$. By the optimality of $p_1(\mathbf{A})_k$,

$$\|p_1(\mathbf{A}) - p_1(\mathbf{A})_k\|_F^2 \leq \|p_1(\mathbf{A}) - p_1(\mathbf{A}_k)\|_F^2 \leq \sum_{i=k+1}^{d} p_1(\sigma_i)^2$$

$$\leq d \cdot \left(\frac{\sigma_{k+1}^2}{2^{2q\sqrt{\epsilon/2}-2}}\right) = O\left(\frac{\epsilon}{2d^2}\sigma_{k+1}^2\right).$$

The last inequalities follow from setting $q = \Theta(\log(d/\epsilon)/\sqrt{\epsilon})$ and from the fact that $\sigma_i \leq \sigma_{k+1} = \alpha$ for all $i \geq k+1$ and thus by property 3 of Lemma 5, $|p_1(\sigma_i)| \leq \frac{\sigma_{k+1}}{2^{q\sqrt{\epsilon/2}-1}}$. Noting that $k \leq d$, we can plug this bound into (4) to get

$$\|p_1(\mathbf{A}) - \mathbf{Y}_1\mathbf{Y}_1^T p_1(\mathbf{A})\|_F^2 \leq \frac{\epsilon}{2}\sigma_{k+1}^2. \tag{5}$$

Applying the Pythagorean theorem and the invariance of the Frobenius norm under rotation gives

$$\|p_1(\mathbf{\Sigma})\|_F^2 - \frac{\epsilon\sigma_{k+1}^2}{2} \leq \|\mathbf{Y}_1\mathbf{Y}_1^T\mathbf{U}p_1(\mathbf{\Sigma})\|_F^2.$$

$\mathbf{Y}_1$ falls within $\mathbf{A}$'s column span, and therefore $\mathbf{U}$'s column span. So we can write $\mathbf{Y}_1 = \mathbf{U}\mathbf{C}$ for some $\mathbf{C} \in \mathbb{R}^{r \times k}$. Since $\mathbf{Y}_1$ and $\mathbf{U}$ have orthonormal columns, so must $\mathbf{C}$. We can now write

$$\|p_1(\mathbf{\Sigma})\|_F^2 - \frac{\epsilon\sigma_{k+1}^2}{2} \leq \|\mathbf{U}\mathbf{C}\mathbf{C}^T\mathbf{U}^T\mathbf{U}p_1(\mathbf{\Sigma})\|_F^2 = \|\mathbf{U}\mathbf{C}\mathbf{C}^T p_1(\mathbf{\Sigma})\|_F^2 = \|\mathbf{C}^T p_1(\mathbf{\Sigma})\|_F^2.$$

Letting $\mathbf{c}_i$ be the $i^{\text{th}}$ row of $\mathbf{C}$, expanding out these norms gives

$$\sum_{i=1}^{r} p_1(\sigma_i)^2 - \frac{\epsilon\sigma_{k+1}^2}{2} \leq \sum_{i=1}^{r} \|\mathbf{c}_i\|_2^2 p_1(\sigma_i)^2. \tag{6}$$

Since $\mathbf{C}$'s columns are orthonormal, its rows all have norms upper bounded by 1. So $\|\mathbf{c}_i\|_2^2 p_1(\sigma_i)^2 \leq p_1(\sigma_i)^2$ for all $i$. So for all $l \leq r$, (6) gives us

$$\sum_{i=1}^{l} (1 - \|\mathbf{c}_i\|_2^2)p_1(\sigma_i)^2 \leq \sum_{i=1}^{r} (1 - \|\mathbf{c}_i\|_2^2)p_1(\sigma_i)^2 \leq \frac{\epsilon\sigma_{k+1}^2}{2}.$$

Recall that $m$ is the number of singular values with $\sigma_i \geq (1 + \epsilon/2)\sigma_{k+1}$. By Property 2 of Lemma 5, for all $i \leq m$ we have $\sigma_i \leq p_1(\sigma_i)$. This gives, for all $l \leq m$:

$$\sum_{i=1}^{l} (1 - \|\mathbf{c}_i\|_2^2)\sigma_i^2 \leq \frac{\epsilon\sigma_{k+1}^2}{2} \text{ and so}$$

$$\sum_{i=1}^{l} \sigma_i^2 - \frac{\epsilon\sigma_{k+1}^2}{2} \leq \sum_{i=1}^{r} \|\mathbf{c}_i\|_2^2\sigma_i^2.$$

Converting these sums back to norms yields $\|\mathbf{\Sigma}_l\|_F^2 - \frac{\epsilon\sigma_{k+1}^2}{2} \leq \|\mathbf{C}^T\mathbf{\Sigma}_l\|_F^2$ and therefore $\|\mathbf{A}_l\|_F^2 - \frac{\epsilon\sigma_{k+1}^2}{2} \leq \|\mathbf{Y}_1\mathbf{Y}_1^T\mathbf{A}_l\|_F^2$ and

$$\|\mathbf{A}_l\|_F^2 - \|\mathbf{Y}_1\mathbf{Y}_1^T\mathbf{A}_l\|_F^2 \leq \frac{\epsilon\sigma_{k+1}^2}{2}. \tag{7}$$

Now $\mathbf{Y}_1\mathbf{Y}_1^T\mathbf{A}_l$ is a rank $l$ approximation to $\mathbf{A}$ falling within the column span of $\mathbf{Y}$ and hence within the column span of $\mathbf{Q}$. By Lemma 2, the best rank $l$ Frobenius approximation to $\mathbf{A}$ within $\mathbf{Q}$ is given by $\mathbf{Q}\bar{\mathbf{U}}_l(\mathbf{Q}\bar{\mathbf{U}}_l)^T\mathbf{A}$. So we have

$$\|\mathbf{A}_l\|_F^2 - \|\mathbf{Q}\bar{\mathbf{U}}_l(\mathbf{Q}\bar{\mathbf{U}}_l)^T\mathbf{A}\|_F^2 = \mathcal{E}(\mathbf{Z}_l,\mathbf{A}) \leq \frac{\epsilon\sigma_{k+1}^2}{2},$$

giving Property 1.

For Algorithm 1, we instead choose $p_1(x) = (1 + \epsilon/2)\sigma_{k+1} \cdot \left(\frac{x}{(1+\epsilon/2)\sigma_{k+1}}\right)^{2q+1}$. For $q = \Theta(\log d/\epsilon)$, this polynomial satisfies the necessary properties: for all $i \geq k+1$, $p_1(\sigma_i) \leq O\left(\frac{\epsilon}{2d^2}\sigma_{k+1}^2\right)$ and for all $i \leq m$, $\sigma_i \leq p_1(\sigma_i)$. Further, up to a rescaling, $p_1(\mathbf{A})\mathbf{\Pi} = \mathbf{K}$ so $\mathbf{Y}_1$ spans the same space as $\mathbf{K}$. Therefore since Algorithm 1 returns $\mathbf{Z}$ with $\mathbf{Z}_l$ equal to the best rank $l$ Frobenius norm approximation to $\mathbf{A}$ within the span of $\mathbf{K}$, for all $l$ we have:

$$\|\mathbf{Q}\bar{\mathbf{U}}_l(\mathbf{Q}\bar{\mathbf{U}}_l)^T\mathbf{A}\|_F^2 \geq \|\mathbf{Y}_1\mathbf{Y}_1^T\mathbf{A}_l\|_F^2 \geq \|\mathbf{A}_l\|_F^2 - \frac{\epsilon\sigma_{k+1}^2}{2},$$

giving the proof.

**Proof of Property 2**

Property 1 and the fact that $\mathcal{E}(\mathbf{Z}_l,\mathbf{A})$ is always positive immediately gives Property 2 for $l \leq m$. So we need to show that it holds for $m < l \leq k$. Note that if $w$, the number of singular values with $\frac{1}{1+\epsilon/2}\sigma_k \leq \sigma_i < \sigma_k$ is equal to 0, then $\sigma_{k+1} < \frac{1}{1+\epsilon/2}\sigma_k$, so $m = k$ and we are done. So we assume $w \geq 1$ henceforth. Again, we first prove the statement for Algorithm 2 and then explain how the proof extends to the simpler case of Algorithm 1.

Intuitively, Property 1 follows from the guarantee that there is a rank $m$ subspace of $span(\mathbf{K})$ that aligns with $\mathbf{A}$ nearly as well as the space spanned by $\mathbf{A}$'s top $m$ singular vectors. To prove Property 2 we must show that there is also some rank $k$ subspace in $span(\mathbf{K})$ whose components all align nearly as well with $\mathbf{A}$ as $\mathbf{u}_k$, the $k^{\text{th}}$ singular vector of $\mathbf{A}$. The existence of such a subspace ensures that $\mathbf{Z}$ performs well, even on singular vectors in the intermediate range $[\sigma_k, (1 + \epsilon/2)\sigma_{k+1}]$.

Let $p_2$ be the polynomial from Lemma 5 with $\alpha = \frac{1}{1+\epsilon/2}\sigma_k$, $\gamma = \epsilon/2$, and $q \geq c\log(d/\epsilon)/\sqrt{\epsilon}$ for some fixed constant $c$. Let $\mathbf{Y}_2 \in \mathbb{R}^{n \times k}$ be an orthonormal basis for the span of $p_2(\mathbf{A})\mathbf{\Pi}$. Again, as long as we choose $q$ to be odd, $p_2(\mathbf{A})$ only contains odd powers of $\mathbf{A}$ and so $\mathbf{Y}_2$ falls within the span of the Krylov subspace from Algorithm 2. We wish to show that for every unit vector $\mathbf{x}$ in the column span of $\mathbf{Y}_2$, $\|\mathbf{x}^T\mathbf{A}\|_2 \geq \frac{1}{1+\epsilon/2}\sigma_k$.

Let $\mathbf{A}_{inner} = \mathbf{A}_{r\backslash k} - \mathbf{A}_{r\backslash(k+w)}$. $\mathbf{A}_{inner} = \mathbf{U}\mathbf{\Sigma}_{inner}\mathbf{V}^T$ where $\mathbf{\Sigma}_{inner}$ contains only the singular values $\sigma_{k+1},\ldots,\sigma_{k+w}$. These are the $w$ intermediate singular values of $\mathbf{A}$ falling in the range $\left[\frac{1}{1+\epsilon/2}\sigma_k, \sigma_k\right)$. Let $\mathbf{A}_{outer} = \mathbf{A} - \mathbf{A}_{inner} = \mathbf{U}\mathbf{\Sigma}_{outer}\mathbf{V}^T$. $\mathbf{\Sigma}_{outer}$ contains all large singular values of $\mathbf{A}$ with $\sigma_i \geq \sigma_k$ and all small singular values with $\sigma_i < \frac{1}{1+\epsilon/2}\sigma_k$.

Let $\mathbf{Y}_{inner} \in \mathbb{R}^{n \times min\{k,w\}}$ be an orthonormal basis for the columns of $p_2(\mathbf{A}_{inner})\mathbf{\Pi}$. Similarly let $\mathbf{Y}_{outer} \in \mathbb{R}^{n \times k,}$ be an orthonormal basis for the columns of $p_2(\mathbf{A}_{outer})\mathbf{\Pi}$.

Every column of $\mathbf{Y}_{inner}$ falls in the column span of $\mathbf{A}_{inner}$ and hence the column span of $\mathbf{U}_{inner} \in \mathbb{R}^{n \times w}$, which contains only the singular vectors of $\mathbf{A}$ corresponding to the inner singular values. Similarly, the columns of $\mathbf{Y}_{outer}$ fall within the span of $\mathbf{U}_{outer} \in \mathbb{R}^{n \times r-w}$, which contains the remaining left singular vectors of $\mathbf{A}$. So the columns of $\mathbf{Y}_{inner}$ are orthogonal to those of $\mathbf{Y}_{outer}$ and $[\mathbf{Y}_{inner},\mathbf{Y}_{outer}]$ forms an orthogonal basis. For any unit vector $\mathbf{x} \in span(p_2(\mathbf{A})\mathbf{\Pi}) = span(\mathbf{Y}_2)$ we can write $\mathbf{x} = \mathbf{x}_{inner} + \mathbf{x}_{outer}$ where $\mathbf{x}_{inner}$ and $\mathbf{x}_{outer}$ are orthogonal vectors in the spans of $\mathbf{Y}_{inner}$ and $\mathbf{Y}_{outer}$ respectively. We have:

$$\|\mathbf{x}^T\mathbf{A}\|_2^2 = \|\mathbf{x}_{inner}^T\mathbf{A}\|_2^2 + \|\mathbf{x}_{outer}^T\mathbf{A}\|_2^2. \tag{8}$$

We will lower bound $\|\mathbf{x}^T\mathbf{A}\|_2^2$ by considering each contribution separately. First, any unit vector $\mathbf{x}' \in \mathbb{R}^n$ in the column span of $\mathbf{Y}_{inner}$ can be written as $\mathbf{x}' = \mathbf{U}_{inner}\mathbf{z}$ where $\mathbf{z} \in \mathbb{R}^w$ is a unit

vector.

$$\|\mathbf{x}'^T\mathbf{A}\|_2^2 = \mathbf{z}^T\mathbf{U}_{inner}^T\mathbf{A}\mathbf{A}^T\mathbf{U}_{inner}\mathbf{z} = \mathbf{z}^T\mathbf{\Sigma}_{inner}^2\mathbf{z} \geq \left(\frac{1}{1+\epsilon/2}\sigma_k\right)^2 \geq (1-\epsilon)\sigma_k^2. \quad (9)$$

Note that we're abusing notation slightly, using $\mathbf{\Sigma}_{inner} \in \mathbb{R}^{w \times w}$ to represent the diagonal matrix containing all singular values of $\mathbf{A}$ with $\frac{1}{1+\epsilon/2}\sigma_k \leq \sigma_i \leq \sigma_k$ without diagonal entries of 0.

We next apply the argument used to prove Property 1 to $p_2(\mathbf{A}_{outer})\mathbf{\Pi}$. The $(k+1)^{\text{th}}$ singular value of $\mathbf{A}_{outer}$ is equal to $\sigma_{k+w+1} \leq \frac{1}{1+\epsilon/2}\sigma_k = \alpha$. So applying (7) we have for all $l \leq k$,

$$\|\mathbf{A}_l\|_F^2 - \|(\mathbf{Y}_{outer})_l (\mathbf{Y}_{outer})_l^T \mathbf{A}_l\|_F^2 \leq \frac{\epsilon\sigma_k^2}{2}. \quad (10)$$

Note that $\mathbf{A}_{outer}$ has the same top $k$ singular vectors at $\mathbf{A}$ so $(\mathbf{A}_{outer})_l = \mathbf{A}_l$. Let $\mathbf{x}' \in \mathbb{R}^n$ be any unit vector within the column space of $\mathbf{Y}_{outer}$ and let $\overline{\mathbf{Y}}_{outer} = (\mathbf{I} - \mathbf{x}'\mathbf{x}'^T)\mathbf{Y}_{outer}$, i.e the matrix with $\mathbf{x}'$ projected off each column. We can use (10) and the optimality of the SVD for low-rank approximation to obtain:

$$\|\mathbf{A}_k\|_F^2 - \|\mathbf{Y}_{outer}\mathbf{Y}_{outer}^T\mathbf{A}_k\|_F^2 \leq \frac{\epsilon\sigma_k^2}{2}$$

$$\|\mathbf{A}_k\|_F^2 - \|\overline{\mathbf{Y}}_{outer}\overline{\mathbf{Y}}_{outer}^T\mathbf{A}_k\|_F^2 - \|\mathbf{x}'\mathbf{x}'^T\mathbf{A}_k\|_F^2 \leq \frac{\epsilon\sigma_k^2}{2}$$

$$\|\mathbf{A}_k\|_F^2 - \|\mathbf{A}_{k-1}\|_F^2 - \frac{\epsilon\sigma_k^2}{2} \leq \|\mathbf{x}'\mathbf{x}'^T\mathbf{A}_k\|_F^2$$

$$(1-\epsilon/2)\sigma_k^2 \leq \|\mathbf{x}'^T\mathbf{A}\|_2^2. \quad (11)$$

Plugging (9) and (11) into (8) yields that, for any $\mathbf{x}$ in $span(\mathbf{Y}_2)$, i.e. $span(p_2(\mathbf{A})\mathbf{\Pi})$,

$$\|\mathbf{x}^T\mathbf{A}\|_2^2 = \|\mathbf{x}_{inner}^T\mathbf{A}\|_2^2 + \|\mathbf{x}_{outer}^T\mathbf{A}\|_2^2$$
$$\geq \left(\|\mathbf{x}_{inner}\|_2^2 + \|\mathbf{x}_{outer}\|_2^2\right)(1-\epsilon)\sigma_k^2 \geq (1-\epsilon)\sigma_k^2. \quad (12)$$

So, we have identified a rank $k$ subspace $\mathbf{Y}_2$ within our Krylov subspace such that every vector in its span aligns at least as well with $\mathbf{A}$ as $\mathbf{u}_k$.

Now, for any $m \leq l \leq k$, consider $\mathcal{E}(\mathbf{Z}_l, \mathbf{A})$. We know that given $\mathbf{Z}_{l-1}$, we can form a rank $l$ matrix $\overline{\mathbf{Z}}_l$ in our Krylov subspace simply by appending a column $\mathbf{x}$ orthogonal to the $l-1$ columns of $\mathbf{Z}_{l-1}$ but falling in the span of $\mathbf{Y}_2$. Since $\mathbf{Y}_2$ has rank $k$, finding such a column is always possible. Since $\mathbf{Z}_l$ is the optimal rank $l$ Frobenius norm approximation to $\mathbf{A}$ falling within our Krylov subspace,

$$\mathcal{E}(\mathbf{Z}_l, \mathbf{A}) \leq \mathcal{E}(\overline{\mathbf{Z}}_l, \mathbf{A}) = \|\mathbf{A}_l\|_F^2 - \|\overline{\mathbf{Z}}_l\overline{\mathbf{Z}}_l^T\mathbf{A}\|_F^2$$
$$= \sigma_l^2 + \|\mathbf{A}_{l-1}\|_F^2 - \|\mathbf{Z}_{l-1}\mathbf{Z}_{l-1}^T\mathbf{A}\|_F^2 - \|\mathbf{x}\mathbf{x}^T\mathbf{A}\|_F^2$$
$$= \mathcal{E}(\mathbf{Z}_{l-1}, \mathbf{A}) + \sigma_l^2 - \|\mathbf{x}\mathbf{x}^T\mathbf{A}\|_F^2$$
$$\leq \mathcal{E}(\mathbf{Z}_{l-1}, \mathbf{A}) + (1+\epsilon/2)^2\sigma_{k+1}^2 - (1-\epsilon)\sigma_{k+1}^2$$
$$\leq \mathcal{E}(\mathbf{Z}_{l-1}, \mathbf{A}) + 3\epsilon \cdot \sigma_{k+1}^2,$$

which gives Property 2.

Again, a nearly identical proof applies for Algorithm 1. We just choose $p_2(x) = \sigma_k\left(\frac{x}{\sigma_k}\right)^{2q+1}$. For $q = \Theta(\log d/\epsilon)$ this polynomial satisfies the necessary properties: for all $i \geq k$, $p_1(\sigma_i) \leq O\left(\frac{\epsilon}{2d^2}\sigma_k^2\right)$ and for all $i \leq k$, $\sigma_i \leq p_2(\sigma_i)$.

**Proof of Property 3**

By Properties 1 and 2 we already have, for all $l \leq k$, $\mathcal{E}(\mathbf{Z}_l, \mathbf{A}) \leq \epsilon\sigma_{k+1}^2 + (l-m) \cdot 3\epsilon\sigma_{k+1}^2 \leq (1+k-m) \cdot 3\epsilon \cdot \sigma_{k+1}^2$. So if $k - m \leq w$ then we immediately have Property 3.

Otherwise, $w < k - m$ so $w < k$ and thus $p_2(\mathbf{A}_{inner})\mathbf{\Pi} \in \mathbb{R}^{n \times k}$ only has rank $w$. It has a null space of dimension $k - w$. Choose any $\mathbf{z}$ in this null space. Then $p_2(\mathbf{A})\mathbf{\Pi}\mathbf{z} = p_2(\mathbf{A}_{inner})\mathbf{\Pi}\mathbf{z} + $

$p_2(\mathbf{A}_{outer})\mathbf{\Pi z} = p_2(\mathbf{A}_{outer})\mathbf{\Pi z}$. In other words, $p_2(\mathbf{A})\mathbf{\Pi z}$ falls entirely within the span of $\mathbf{Y}_{outer}$. So, there is a $k - w$ dimensional subspace of $span(\mathbf{Y}_2)$ that is entirely contained in $span(\mathbf{Y}_{outer})$.

For $l \leq m + w$, then Properties 1 and 2 already give us $\mathcal{E}(\mathbf{Z}_l, \mathbf{A}) \leq \epsilon\sigma_{k+1}^2 + (l - m) \cdot 3\epsilon\sigma_{k+1}^2 \leq (w+1) \cdot 3\epsilon \cdot \sigma_{k+1}^2$. So consider $m + w \leq l \leq k$. Given $\mathbf{Z}_m$, to form a rank $l$ matrix $\overline{\mathbf{Z}}_l$ in our Krylov subspace we need to append $l - m$ orthonormal columns. We can choose $\min\{k - w - m, l - m\}$ columns, $\mathbf{X}_1$, from the $k - w$ dimensional subspace within $span(\mathbf{Y}_2)$ that is entirely contained in $span(\mathbf{Y}_{outer})$. If necessary (i.e. $k - w - m \leq l - m$), We can then choose the remaining $l - (k - w)$ columns $\mathbf{X}_2$ from the span of $\mathbf{Y}_2$.

Similar to our argument when considering a single vector in the span of $\mathbf{Y}_{outer}$, letting $\overline{\mathbf{Y}}_{outer} = \left(\mathbf{I} - \mathbf{X}_1\mathbf{X}_1^T\right)\mathbf{Y}_{outer}$, we have by (10):

$$\|\mathbf{A}_k\|_F^2 - \|\mathbf{Y}_{outer}\mathbf{Y}_{outer}^T\mathbf{A}_k\|_F^2 \leq \frac{\epsilon\sigma_k^2}{2}$$

$$\|\mathbf{A}_k\|_F^2 - \|\overline{\mathbf{Y}}_{outer}\overline{\mathbf{Y}}_{outer}^T\mathbf{A}_k\|_F^2 - \|\mathbf{X}_1\mathbf{X}_1^T\mathbf{A}_k\|_F^2 \leq \frac{\epsilon\sigma_k^2}{2}$$

$$\|\mathbf{A}_k\|_F^2 - \|\mathbf{A}_{k-\min\{k-w-m,l-m\}}\|_F^2 - \frac{\epsilon\sigma_k^2}{2} \leq \|\mathbf{X}_1\mathbf{X}_1^T\mathbf{A}_k\|_F^2$$

$$\sum_{i=k-\min\{k-w-m,l-m\}+1}^{k} \sigma_i^2 - \frac{\epsilon\sigma_k^2}{2} \leq \|\mathbf{X}_1\mathbf{X}_1^T\mathbf{A}\|_F^2.$$

By applying (12) directly to each column of $\mathbf{X}_2$ we also have:

$$(l + w - k)\sigma_k^2 - (l + w - k)\epsilon\sigma_k^2 \leq \|\mathbf{X}_2\mathbf{X}_2^T\mathbf{A}\|_F^2$$
$$(l + w - k)\sigma_{k+1}^2 - (l + w - k)\epsilon\sigma_{k+1}^2 \leq \|\mathbf{X}_2\mathbf{X}_2^T\mathbf{A}\|_F^2.$$

Assume that $\min\{k - w - m, l - m\} = k - w - m$. Similar calculations show the same result when $\min\{k - w - m, l - m\} = l - m$. We can use the above two bounds to obtain:

$$\mathcal{E}(\mathbf{Z}_l, \mathbf{A}) \leq \mathcal{E}(\overline{\mathbf{Z}}_l, \mathbf{A})$$

$$= \|\mathbf{A}_l\|_F^2 - \|\overline{\mathbf{Z}}_l\overline{\mathbf{Z}}_l^T\mathbf{A}\|_F^2$$

$$= \sum_{i=m+1}^{l} \sigma_i^2 + \|\mathbf{A}_m\|_F^2 - \|\mathbf{Z}_m\mathbf{Z}_m^T\mathbf{A}\|_F^2 - \|\mathbf{X}_1\mathbf{X}_1^T\mathbf{A}\|_F^2 - \|\mathbf{X}_2\mathbf{X}_2^T\mathbf{A}\|_F^2$$

$$\leq \mathcal{E}(\mathbf{Z}_m, \mathbf{A}) + \sum_{i=m+1}^{l} \sigma_i^2 - \sum_{i=w+m+1}^{k} \sigma_i^2 + \frac{\epsilon\sigma_k^2}{2} - (l + w - k)\sigma_{k+1}^2 + (l + w - k)\epsilon\sigma_{k+1}^2$$

$$\leq \sum_{i=m+1}^{m+w} \sigma_i^2 - w\sigma_{k+1}^2 + (l + w - k + 3/2)\epsilon\sigma_{k+1}^2$$

$$\leq (l + 3w - k + 3/2)\epsilon\sigma_{k+1}^2$$

$$\leq (w + 1) \cdot 3\epsilon \cdot \sigma_{k+1}^2,$$

giving Property 3 for all $l \leq k$. $\qquad\square$

## 6.2 Error Bounds for Simultaneous Iteration and Block Krylov Iteration

With Lemma 9 in place, we can easily prove that Simultaneous Iteration and Block Krylov Iteration both achieve the low-rank approximation and PCA guarantees (1), (2), and (3).

**Theorem 10** (Near Optimal Spectral Norm Error Approximation). *With probability* 99/100, *Algorithms 1 and 2 return* $\mathbf{Z}$ *satisfying* (2):

$$\|\mathbf{A} - \mathbf{Z}\mathbf{Z}^T\mathbf{A}\|_2 \leq (1 + \epsilon)\|\mathbf{A} - \mathbf{A}_k\|_2.$$

*Proof.* Let $m$ be the number of singular values with $\sigma_i \geq (1+\epsilon/2)\sigma_{k+1}$. If $m = 0$ then we are done since any $\mathbf{Z}$ will satisfy $\|\mathbf{A} - \mathbf{ZZ}^T\mathbf{A}\|_2 \leq \|\mathbf{A}\|_2 = \sigma_1 \leq (1+\epsilon/2)\sigma_{k+1} \leq (1+\epsilon)\|\mathbf{A} - \mathbf{A}_k\|_2$. Otherwise, by Property 1 of Lemma 9,

$$\mathcal{E}(\mathbf{Z}_m, \mathbf{A}) \leq \frac{\epsilon\sigma_{k+1}^2}{2}$$

$$\|\mathbf{A} - \mathbf{Z}_m\mathbf{Z}_m^T\mathbf{A}\|_F^2 \leq \|\mathbf{A} - \mathbf{A}_m\|_F^2 + \frac{\epsilon\sigma_{k+1}^2}{2}.$$

Additive error in Frobenius norm directly translates to additive spectral norm error. Specifically, applying Theorem 3.4 of [22], which we also prove as Lemma 15 in Appendix A,

$$\|\mathbf{A} - \mathbf{Z}_m\mathbf{Z}_m^T\mathbf{A}\|_2^2 \leq \|\mathbf{A} - \mathbf{A}_m\|_2^2 + \frac{\epsilon\sigma_{k+1}^2}{2} \leq \sigma_{m+1}^2 + \frac{\epsilon\sigma_{k+1}^2}{2}$$

$$\leq (1+\epsilon/2)\sigma_{k+1}^2 + \frac{\epsilon\sigma_{k+1}^2}{2} \leq (1+\epsilon)\|\mathbf{A} - \mathbf{A}_k\|_2^2. \tag{13}$$

Finally, $\mathbf{Z}_m\mathbf{Z}_m^T\mathbf{A} = \mathbf{ZZ}_m^T\mathbf{A}$ and so by Lemma 3 we have $\|\mathbf{A} - \mathbf{ZZ}^T\mathbf{A}\|_2^2 \leq \|\mathbf{A} - \mathbf{Z}_m\mathbf{Z}_m^T\mathbf{A}\|_2^2$, which combines with (13) to give the result. $\qquad\square$

**Theorem 11** (Near Optimal Frobenius Norm Error Approximation). *With probability* $99/100$*, Algorithms 1 and 2 return* $\mathbf{Z}$ *satisfying* (1)*:*

$$\|\mathbf{A} - \mathbf{ZZ}^T\mathbf{A}\|_F \leq (1+\epsilon)\|\mathbf{A} - \mathbf{A}_k\|_F.$$

*Proof.* By Property 3 of Lemma 9 we have:

$$\mathcal{E}(\mathbf{Z}_l, \mathbf{A}) \leq (w+1) \cdot 3\epsilon \cdot \sigma_{k+1}^2$$
$$\|\mathbf{A} - \mathbf{ZZ}^T\mathbf{A}\|_F^2 \leq \|\mathbf{A} - \mathbf{A}_k\|_F^2 + (w+1) \cdot 3\epsilon \cdot \sigma_{k+1}^2. \tag{14}$$

$w$ is defined as the number of singular values with $\frac{1}{1+\epsilon/2}\sigma_k \leq \sigma_i < \sigma_k$. So $\|\mathbf{A} - \mathbf{A}_k\|_F^2 \geq w \cdot \left(\frac{1}{1+\epsilon/2}\sigma_k\right)^2$. Plugging into (14) we have:

$$\|\mathbf{A} - \mathbf{ZZ}^T\mathbf{A}\|_F^2 \leq \|\mathbf{A} - \mathbf{A}_k\|_F^2 + (w+1) \cdot 3\epsilon \cdot \sigma_{k+1}^2 \leq (1+10\epsilon)\|\mathbf{A} - \mathbf{A}_k\|_F^2.$$

Adjusting constants on the $\epsilon$ gives us the result. $\qquad\square$

**Theorem 12** (Per Vector Quality Guarantee). *With probability* $99/100$*, Algorithms 1 and 2 return* $\mathbf{Z}$ *satisfying* (3)*:*

$$\forall i, \; \left|\mathbf{u}_i^T\mathbf{AA}^T\mathbf{u}_i - \mathbf{z}_i^T\mathbf{AA}^T\mathbf{z}_i\right| \leq \epsilon\sigma_{k+1}^2.$$

*Proof.* First note that $\mathbf{z}_i^T\mathbf{AA}^T\mathbf{z}_i \leq \mathbf{u}_i^T\mathbf{AA}^T\mathbf{u}_i$. This is because $\mathbf{z}_i^T\mathbf{AA}^T\mathbf{z}_i = \mathbf{z}_i^T\mathbf{QQ}^T\mathbf{AA}^T\mathbf{QQ}^T\mathbf{z}_i = \sigma_i(\mathbf{QQ}^T\mathbf{A})^2$ by our choice of $\mathbf{z}_i$. $\sigma_i(\mathbf{QQ}^T\mathbf{A})^2 \leq \sigma_i(\mathbf{A})^2$ since applying a projection to $\mathbf{A}$ will decrease each of its singular values (which follows for example from the Courant-Fischer min-max principle). Then by Property 2 of Lemma 9 we have, for all $i \leq k$,

$$\|\mathbf{A}_i\|_F^2 - \|\mathbf{Z}_i\mathbf{Z}_i^T\|_F^2 \leq \|\mathbf{A}_{i-1}\|_F^2 - \|\mathbf{Z}_{i-1}\mathbf{Z}_{i-1}^T\|_F^2 + 3\epsilon\sigma_{k+1}^2$$
$$\sigma_i^2 \leq \|\mathbf{z}_i\mathbf{z}_i^T\mathbf{A}\|_F^2 + 3\epsilon\sigma_{k+1}^2 = \mathbf{z}_i^T\mathbf{AA}^T\mathbf{z}_i + 3\epsilon\sigma_{k+1}^2.$$

$\sigma_i^2 = \mathbf{u}_i^T\mathbf{AA}^T\mathbf{u}_i$, so simply adjusting constants on $\epsilon$ gives the result. $\qquad\square$

# 7 Improved Convergence With Spectral Decay

In addition to the implementations of Simultaneous Iteration and Block Krylov Iteration given in Algorithms 1 and 2, our analysis applies to the common modification of running the algorithms with $\mathbf{\Pi} \in \mathbb{R}^{n \times p}$ for $p \geq k$ [1, 20, 2]. This technique can significantly accelerate both methods for matrices with decaying singular values. For simplicity, we focus on Block Krylov Iteration, although as usual all arguments immediately extend to the simpler Simultaneous Iteration algorithm.

In order to avoid inverse dependence on the potentially small singular value gap $\frac{\sigma_k}{\sigma_{k+1}} - 1$, the number of Block Krylov iterations inherently depends on $1/\sqrt{\epsilon}$. This ensures that our matrix polynomial sufficiently separates small singular values from larger ones. However, when $\sigma_k > (1 + \epsilon)\sigma_{k+1}$ we can actually use $q = \Theta\left(\log(d/\epsilon)/\sqrt{\min\{1, \frac{\sigma_k}{\sigma_{k+1}} - 1\}}\right)$ iterations, which is sufficient for separating the top $k$ singular values significantly from the lower values. Specifically, if we set $\alpha = \sigma_{k+1}$ and $\gamma = \frac{\sigma_k}{\sigma_{k+1}} - 1$, we know that with $q = \Theta\left(\log(d/\epsilon)/\sqrt{\min\{1, \frac{\sigma_k}{\sigma_{k+1}} - 1\}}\right)$, (5) still holds. We can then just follow the proof of Lemma 9 and show that Property 1 holds for *all* $l \leq k$ (not just for $l \leq m$ as originally proven). This gives Property 2 and Property 3 trivially.

Further, for $p \geq k$, the exact same analysis shows that $q = \Theta\left(\log(d/\epsilon)/\sqrt{\min\{1, \frac{\sigma_k}{\sigma_{p+1}} - 1\}}\right)$ suffices. When $\mathbf{A}$'s spectrum decays rapidly, so $\sigma_{p+1} \leq c \cdot \sigma_k$ for some constant $c < 1$ and some $p$ not much larger than $k$, we can obtain significantly faster runtimes. Our $\epsilon$ dependence becomes logarithmic, rather than polynomial:

**Theorem 13** (Gap Dependent Convergence). *With probability* $99/100$*, for any* $p \geq k$*, Algorithm 1 or 2 initialized with* $\mathbf{\Pi} \sim \mathcal{N}(0,1)^{d \times p}$ *returns* $\mathbf{Z}$ *satisfying guarantees* (1)*,* (2)*, and* (3) *as long as we set* $q = \Theta\left(\log(d/\epsilon)/\left(\min\{1, \frac{\sigma_k}{\sigma_{p+1}} - 1\}\right)\right)$ *or* $\Theta\left(\log(d/\epsilon)/\sqrt{\min\{1, \frac{\sigma_k}{\sigma_{p+1}} - 1\}}\right)$*, respectively.*

This theorem may prove especially useful in practice because, on many architectures, multiplying a large $\mathbf{A}$ by $2k$ or even $10k$ vectors is not much more expensive than multiplying by $k$ vectors. Additionally, it should still be possible to perform all steps for post-processing $\mathbf{K}$ in memory, again limiting additional runtime costs due to its larger size.

Finally, we note that while Theorem 13 is more reminiscent of classical gap-dependent bounds, it still takes substantial advantage of the fact that we're looking for nearly optimal low-rank approximations and principal components instead of attempting to converge precisely to $\mathbf{A}$'s true singular vectors. This allows the result to avoid dependence on the gap between *adjacent* singular values, instead varying only with $\frac{\sigma_k}{\sigma_{p+1}}$, which should be much larger.

## 8  Experiments

We close with several experimental results. A variety of empirical papers, not to mention widespread adoption, already justify the use of randomized SVD algorithms. Prior work focuses in particular on benchmarking Simultaneous Iteration [20, 12] and, due to its improved accuracy over sketch-and-solve approaches, this algorithm is popular in practice [11, 17]. As such, we focus on demonstrating that for many data problems Block Krylov Iteration can offer significantly better convergence.

We implement both algorithms in MATLAB using Gaussian random starting matrices with exactly $k$ columns. We explicitly compute $\mathbf{K}$ for both algorithms, as described in Section 5, and use re-orthonormalization at each iteration to improve stability [45]. We test the algorithms with varying iteration count $q$ on three common datasets, SNAP/AMAZON0302 [23, 24], SNAP/EMAIL-ENRON [23, 46], and 20 NEWSGROUPS [25], computing column principal components in all cases. We plot error vs. iteration count for metrics (1), (2), and (3) in Figure 3. For per vector error (3), we plot the maximum deviation amongst all top $k$ approximate principal components (relative to $\sigma_{k+1}$).

Unsurprisingly, both algorithms obtain very accurate Frobenius norm error, $\|\mathbf{A} - \mathbf{Z}\mathbf{Z}^T\mathbf{A}\|_F/\|\mathbf{A} - \mathbf{A}_k\|_F$, with very few iterations. This is our intuitively weakest guarantee and, in the presence of a heavy singular value tail, both iterative algorithms will outperform the worst case analysis.

On the other hand, for spectral norm low-rank approximation and per vector error, we confirm that Block Krylov Iteration converges much more rapidly than Simultaneous Iteration, as predicted by our theoretical analysis. It it often possible to achieve nearly optimal error with $< 8$ iterations where as getting to within say $1\%$ error with Simultaneous Iteration can take much longer.

The final plot in Figure 3 shows error verses runtime for the $11269 \times 15088$ dimensional 20 NEWSGROUPS dataset. We averaged over 7 trials and ran the experiments on a commodity laptop with 16GB of memory. As predicted, because its additional memory overhead and post-processing costs

(a) SNAP/AMAZON0302, $k = 30$

(b) SNAP/EMAIL-ENRON, $k = 10$

(c) 20 NEWSGROUPS, $k = 20$

(d) 20 NEWSGROUPS, $k = 20$, runtime cost

Figure 3: Low-rank approximation and per vector error convergence rates for Algorithms 1 and 2.

are small compared to the cost of the large matrix multiplication required for each iteration, Block Krylov Iteration outperforms Simultaneous Iteration for small $\epsilon$.

More generally, these results justify the importance of convergence bounds that are independent of singular value gaps. Our analysis in Section 7 predicts that, once $\epsilon$ is small in comparison to the gap $\frac{\sigma_k}{\sigma_{k+1}} - 1$, we should see much more rapid convergence since $q$ will depend on $\log(1/\epsilon)$ instead of $1/\epsilon$. However, for Simultaneous Iteration, we do not see this behavior with SNAP/AMAZON0302 and it only just begins to emerge for 20 NEWSGROUPS.

While all three datasets have rapid singular value decay, a careful look confirms that their singular value gaps are actually quite small! For example, $\frac{\sigma_k}{\sigma_{k+1}} - 1$ is .004 for SNAP/AMAZON0302 and .011 for 20 NEWSGROUPS, in comparison to .042 for SNAP/EMAIL-ENRON. Accordingly, the frequent claim that singular value gaps can be taken as constant is insufficient, even for small $\epsilon$.

## Acknowledgments

We thank David Woodruff, Aaron Sidford, Richard Peng and Jon Kelner for several valuable conversations. Additionally, Michael Cohen was very helpful in discussing many details of this project, including the ultimate form of Lemma 9. This work was partially supported by NSF Graduate Research Fellowship Grant No. 1122374, AFOSR grant FA9550-13-1-0042, DARPA grant FA8650-11-C-7192, and the NSF Center for Science of Information.

## Footnotes

[1]Typically after mean centering $\mathbf{A}$'s columns or rows, depending on which principal components we want.

[2]This is somewhat of an oversimplicifcation. By the Abel-Ruffini Theorem, an *exact* SVD is incomputable even with exact arithmetic [3]. Accordingly, all SVD algorithm are inherently iteratively. Nevertheless, traditional methods including the ubiquitous QR algorithm obtain superlinear convergence rates for the low-rank approximation problem. In any reasonable computing environment, they can be taken to run in $O(nd^2)$ time.

[3]Here $\text{nnz}(\mathbf{A})$ is the number of non-zero entries in $\mathbf{A}$ and this runtime hides lower order terms.

[4]In fact, it does not even imply $(1 + \epsilon)$ Frobenius norm error.

[5]For guarantee (3) it is important that Algorithm 1 includes post-processing steps 4 and 5 rather than just returning a basis for $\mathbf{K}$, which is sufficient for the low-rank approximation guarantees.

[6]For nonsymmetric matrices we work with $(\mathbf{AA}^T)^q\mathbf{A}$, but present the symmetric case here for simplicity.

[7]Algorithm 2 in fact only constructs odd powered terms in $\mathbf{K}$, which is sufficient for our choice of $p_q(x)$.

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

# A  Appendix

**Frobenius Norm Low-Rank Approximation**

We first give a deterministic Lemma, from which the main approximation result follows.

**Lemma 14** (Special case of Lemma 4.4 of [14], originally proven in [16])**.** *Let* $\mathbf{A} \in \mathbb{R}^{n \times d}$ *have SVD* $\mathbf{A} = \mathbf{U}\mathbf{\Sigma}\mathbf{V}^T$*, let* $\mathbf{S} \in \mathbb{R}^{d \times k}$ *be any matrix such that* $\mathrm{rank}\left(\mathbf{V}_k^T \mathbf{S}\right) = k$*, and let* $\mathbf{C} \in \mathbb{R}^{n \times k}$ *be an orthonormal basis for the column span of* $\mathbf{AS}$*. Then:*

$$\|\mathbf{A} - \mathbf{C}\mathbf{C}^T\mathbf{A}\|_F^2 \leq \|\mathbf{A} - \mathbf{A}_k\|_F^2 + \|\left(\mathbf{A} - \mathbf{A}_k\right)\mathbf{S}\left(\mathbf{V}_k^T\mathbf{S}\right)^+\|_F^2.$$

**Lemma 4** (Frobenius Norm Low-Rank Approximation). *For any $\mathbf{A} \in \mathbb{R}^{n \times d}$ and $\mathbf{\Pi} \in \mathbb{R}^{d \times k}$ where the entries of $\mathbf{\Pi}$ are independent Gaussians drawn from $\mathcal{N}(0,1)$. If we let $\mathbf{Z}$ be an orthonormal basis for $span\left(\mathbf{A\Pi}\right)$, then with probability at least $99/100$, for some fixed constant $c$,*

$$\|\mathbf{A} - \mathbf{Z}\mathbf{Z}^T\mathbf{A}\|_F^2 \leq c \cdot dk\|\mathbf{A} - \mathbf{A}_k\|_F^2.$$

*Proof.* We follow [14]. Apply Lemma 14 with $\mathbf{S} = \mathbf{\Pi}$. With probability 1, $\mathbf{V}_k^T\mathbf{S}$ has full rank. So, to show the result we need to show that $\|\left(\mathbf{A} - \mathbf{A}_k\right)\mathbf{S}\left(\mathbf{V}_k^T\mathbf{S}\right)^+\|_F^2 \leq c\|\mathbf{A} - \mathbf{A}_k\|_F^2$ for some fixed $c$. For any two matrices $\mathbf{M}$ and $\mathbf{N}$, $\|\mathbf{M}\mathbf{N}\|_F \leq \|\mathbf{M}\|_F\|\mathbf{N}\|_2$. This property is known as *spectral submultiplicativity*. Noting that $\|\mathbf{U}_{r\backslash k}\mathbf{\Sigma}_{r\backslash k}\|_F^2 = \|\mathbf{A} - \mathbf{A}_k\|_F^2$ and applying submultiplicativity,

$$\|\left(\mathbf{A} - \mathbf{A}_k\right)\mathbf{S}\left(\mathbf{V}_k^T\mathbf{S}\right)^+\|_F^2 \leq \|\mathbf{U}_{r\backslash k}\mathbf{\Sigma}_{r\backslash k}\|_F^2\|\mathbf{V}_{r\backslash k}^T\mathbf{S}\|_2^2\|\left(\mathbf{V}_k^T\mathbf{S}\right)^+\|_2^2.$$

By the rotational invariance of the Gaussian distribution, since the rows of $\mathbf{V}^T$ are orthonormal, the entries of $\mathbf{V}_k^T\mathbf{S}$ and $\mathbf{V}_{r\backslash k}^T\mathbf{S}$ are independent Gaussians. By standard Gaussian matrix concentration results (Fact 6 of [14], also in [47]), with probability at least $99/100$, $\|\mathbf{V}_{r\backslash k}^T\mathbf{S}\|_2^2 \leq c_1 \cdot \max\{k, r - k\} \leq c_1\dot{d}$ and $\|\left(\mathbf{V}_k^T\mathbf{S}\right)^+\|_2^2 \leq c_2 k$ for some fixed constants $c_1, c_2$. So,

$$\|\mathbf{U}_{r\backslash k}\mathbf{\Sigma}_{r\backslash k}\|_F^2\|\mathbf{V}_{r\backslash k}^T\mathbf{S}\|_2^2\|\left(\mathbf{V}_k^T\mathbf{S}\right)^+\|_2^2 \leq c \cdot dk\|\mathbf{A} - \mathbf{A}_k\|_F^2$$

for some fixed $c$, yielding the result. Note that we choose probability $99/100$ for simplicity – we can obtain a result with higher probability by simply allowing for a higher constant $c$, which in our applications of Lemma 4 will only factor into logarithmic terms. $\quad\square$

### Chebyshev Polynomials

**Lemma 5** (Chebyshev Minimizing Polynomial). *Given a specified value $\alpha > 0$, gap $\gamma \in (0,1]$, and $q \geq 1$, there exists a degree $q$ polynomial $p(x)$ such that:*

1. $p((1 + \gamma)\alpha) = (1 + \gamma)\alpha$

2. $p(x) \geq x$ *for all* $x \geq (1 + \gamma)\alpha$

3. $|p(x)| \leq \frac{\alpha}{2^{q\sqrt{\gamma} - 1}}$ *for all* $x \in [0, \alpha]$

*Furthermore, when $q$ is odd, the polynomial only contains odd powered monomials.*

*Proof.* The required polynomial can be constructed using a standard Chebyshev polynomial of degree $q$, $T_q(x)$, which is defined by the three term recurrence:

$$T_0(x) = 1$$
$$T_1(x) = x$$
$$T_q(x) = 2xT_{q-1}(x) - T_{q-2}(x)$$

Each Chebyshev polynomial satisfies the well known property that $T_q(x) \leq 1$ for all $x \in [-1, 1]$ and, for $x > 1$, we can write the polynomials in closed form [48]:

$$T_q(x) = \frac{(x + \sqrt{x^2 - 1})^q + (x - \sqrt{x^2 - 1})^q}{2}. \tag{15}$$

For Lemma 5, we simply set:

$$p(x) = (1 + \gamma)\alpha\frac{T_q(x/\alpha)}{T_q(1 + \gamma)}, \tag{16}$$

which is clearly of degree $q$ and well defined since, referring to (15), $T_q(x) > 0$ for all $x > 1$. Now,

$$p((1 + \gamma)\alpha) = (1 + \gamma)\alpha\frac{T_q(1 + \gamma)}{T_q(1 + \gamma)} = (1 + \gamma)\alpha,$$

so $p(x)$ satisfies property 1. With property 1 in place, to prove that $p(x)$ satisfies property 2, it suffices to show that $p'(x) \geq 1$ for all $x \geq (1 + \gamma)\alpha$. By chain rule,

$$p'(x) = \frac{(1 + \gamma)}{T_q(1 + \gamma)}T_q'(x/\alpha).$$

Thus, it suffices to prove that, for all $x \geq (1 + \gamma)$,

$$(1 + \gamma)T_q'(x) \geq T_q(1 + \gamma). \tag{17}$$

We do this by showing that $(1+\gamma)T_q'(1+\gamma) \geq T_q(1+\gamma)$ and then claim that $T_q''(x) \geq 0$ for all $x > (1+\gamma)$, so (17) holds for $x > (1 + \gamma)$ as well. A standard form for the derivative of the Chebyshev polynomial is

$$T_q' = \begin{cases} 2q\left(T_{q-1} + T_{q-3} + \ldots + T_1\right) & \text{if } q \text{ is even,} \\ 2q\left(T_{q-1} + T_{q-3} + \ldots + T_2\right) + q & \text{if } q \text{ is odd.} \end{cases} \tag{18}$$

(18) can be verified via induction once noting that the Chebyshev recurrence gives $T_q' = 2xT_{q-1}' + 2T_{q-1} - T_{q-2}'$. Since $T_i(x) > 0$ when $x \geq 1$, we can conclude that $T_q'(x) \geq 2qT_{q-1}(x)$. So proving (17) for $x = (1 + \gamma)$ reduces to proving that

$$(1 + \gamma)2qT_{q-1}(1 + \gamma) \geq T_q(1 + \gamma). \tag{19}$$

Noting that, for $x \geq 1$, $(x + \sqrt{x^2 - 1}) > 0$ and $(x - \sqrt{x^2 - 1}) > 0$, it follows from (15) that

$$T_{q-1}(x)\left((x + \sqrt{x^2 - 1}) + (x - \sqrt{x^2 - 1})\right) \geq T_q(x),$$

and thus

$$\frac{T_q(x)}{T_{q-1}(x)} \leq 2x.$$

So, to prove (19), it suffices to show that $2(1 + \gamma) \leq (1 + \gamma)2q$, which is true whenever $q \geq 1$. So (17) holds for all $x = (1 + \gamma)$.

Finally, referring to (18), we know that $T_q''$ must be some positive combination of lower degree Chebyshev polynomials. Again, since $T_i(x) > 0$ when $x \geq 1$, we conclude that $T_q''(x) \geq 0$ for all $x \geq 1$. It follows that $T_q'(x)$ does not decrease above $x = (1 + \gamma)$, so (17) also holds for all $x > (1 + \gamma)$ and we have proved property 2.

To prove property 3, we first note that, by the well known property that $T_i(x) \leq 1$ for $x \in [-1, 1]$, $T_q(x/\alpha) \leq 1$ for $x \in [0, \alpha]$. So, to prove $p(x) \leq \frac{\alpha}{2^{q\sqrt{\gamma}-1}}$, we just need to show that

$$\frac{1}{T_q(1 + \gamma)} \leq \frac{1}{2^{q\sqrt{\gamma}-1}}. \tag{20}$$

Equation (15) gives $T_q(1+\gamma) \geq \frac{1}{2}(1+\gamma+\sqrt{(1 + \gamma)^2 - 1})^q \geq \frac{1}{2}(1+\sqrt{\gamma})^q$. When $\gamma \leq 1$, $(1+\sqrt{\gamma})^{1/\sqrt{\gamma}} \geq 2$. Thus, $(1 + \sqrt{\gamma})^q \geq 2^{q\sqrt{\gamma}}$. Dividing by 2 gives $T_q(1 + \gamma) \geq 2^{q\sqrt{\gamma}-1}$, which gives (20) and thus property 3.

Finally, we remark that it is well known that odd degree Chebyshev polynomials of the first kind only contain monomials of odd degree (and this is easy to verify inductively). Accordingly, since $p_q(x)$ is simply a scaling of $T_q(x)$, if we choose $q$ to be odd, $p_q(x)$ only contains odd degree terms. □

**Additive Frobenius Norm Error Implies Additive Spectral Norm Error**

**Lemma 15** (Theorem 3.4 of [22]). *For any $\mathbf{A} \in \mathbb{R}^{n \times d}$, let $\mathbf{B} \in \mathbb{R}^{n \times d}$ be any rank $k$ matrix satisfying $\|\mathbf{A} - \mathbf{B}\|_F^2 \leq \|\mathbf{A} - \mathbf{A}_k\|_F^2 + \eta$. Then*

$$\|\mathbf{A} - \mathbf{B}\|_2^2 \leq \|\mathbf{A} - \mathbf{A}_k\|_2^2 + \eta.$$

*Proof.* We follow the proof given in [22] nearly exactly, including it for completeness. By Weyl's monotonicity theorem (Theorem 3.2 in [22]), for any two matrices $\mathbf{X}, \mathbf{Y} \in \mathbb{R}^{n \times d}$ with $n \geq d$, for all $i, j$ with $i + j - 1 \leq n$ we have $\sigma_{i+j-1}(\mathbf{X} + \mathbf{Y}) \leq \sigma_i(\mathbf{X}) + \sigma_j(\mathbf{X})$. If we write $\mathbf{A} = (\mathbf{A} - \mathbf{B}) + \mathbf{B}$ and apply this theorem, then for all $1 \geq i \geq n - k$,

$$\sigma_{i+k}(\mathbf{A}) \leq \sigma_i(\mathbf{A} - \mathbf{B}) + \sigma_{k+1}(\mathbf{B}).$$

Note that if $n < d$, we can just work with $\mathbf{A}^T$ and $\mathbf{B}^T$. Now, $\sigma_{k+1}(\mathbf{B}) = 0$ since $\mathbf{B}$ is rank $k$. Using the resulting inequality and recalling that $\|\mathbf{A} - \mathbf{A}_k\|_F^2 = \sum_{i=k+1}^n \sigma_i^2(\mathbf{A})$, we see that:

$$\|\mathbf{A} - \mathbf{B}\|_F^2 \leq \|\mathbf{A} - \mathbf{A}_k\|_F^2 + \eta$$

$$\sum_{i=1}^n \sigma_i^2(\mathbf{A} - \mathbf{B}) \leq \sum_{i=k+1}^n \sigma_i^2(\mathbf{A}) + \eta$$

$$\sum_{i=1}^{n-k} \sigma_i^2(\mathbf{A} - \mathbf{B}) \leq \sum_{i=k+1}^n \sigma_i^2(\mathbf{A}) + \eta$$

$$\sigma_1^2(\mathbf{A} - \mathbf{B}) + \sum_{i=2}^{n-k} \sigma_i^2(\mathbf{A}) \leq \sum_{i=k+1}^n \sigma_i^2(\mathbf{A}) + \eta$$

$$\sigma_1^2(\mathbf{A} - \mathbf{B}) \leq \sum_{i=k+1}^n \sigma_i^2(\mathbf{A}) - \sum_{i=2}^{n-k} \sigma_i^2(\mathbf{A}) + \eta$$

$$\sigma_1^2(\mathbf{A} - \mathbf{B}) \leq \sigma_{k+1}^2(\mathbf{A}) + \eta.$$

$\sigma_{k+1}^2(\mathbf{A})$ is equal to the squared top singular value of $\mathbf{A} - \mathbf{A}_k$ (i.e. $\|\mathbf{A} - \mathbf{A}_k\|_2^2$, so the lemma follows. $\qquad\square$