[Reviews · NeurIPS 2015]

Submitted by Assigned_Reviewer_1

The authors may want to relate their work to that of "Estimating the Largest Eigenvalue by the Power and Lanczos Algorithms with a Random Start", Kuczynski, J. and Wozniakowski, H., 1992.
Summary: The authors analyze the block Lanczos algorithm from a randomized numerical linear algebra point of view, and confirm that as in the deterministic setting, Lanczos is far superior to block power method. The paper is well written and highly relevant.

Submitted by Assigned_Reviewer_2

The authors discuss SVD approximation using a (randomized) Block Lanczos algorithm. The idea for the algorithm is not new (has been proposed before) but the author's claim of proven convergence guarantee is better than the (state-of-the-art) guarantee for the simultaneous power iteration.

The proofs are relegated to the full version of the paper, and while that would not prevent acceptance (in the minds of this reviewer), when the major contribution is in these proofs (with concerns regarding the experiment, applicability, see later) then this makes the paper less suitable for NIPS.

One compelling insight is that to compute a low rank approximation (classically done via a partial SVD) one may not necessarily need to produce exact (and orthogonal etc.) singular vectors (and then form the low rank approximator) and therefore the (numerical, orthogonality) issues arising from close singular values may not prevent the finding of a low rank approximation itself efficiently (and for any matrix). However, this is not how the authors communicated their message, case in point: low rank approximation does not get mentioned in the title and only indirectly in the abstract when referring very briefly to PCA.

It is well known that the Lanczos (block Lanczos) algorithm suffers from numerical issues manifesting itself in losing orthogonality (in floating point arithmetic) that is guaranteed in exact arithmetic. A comprehensive read on this topic is "Lanczos Algorithms for Large Symmetric Eigenvalue Computations" by J.K. Cullum and R.A Willoughby (SIAM classics in applied mathematics, 41). Various degrees of reorthogonalization have been considered to address this issue, which is of course expensive and limit the number of eigenvalues that can be computed to the required accuracy. Paige in 1980 have shown what exactly is behind the loss of accuracy and it is not simply the result of roundoff error but the interaction between roundoff and convergence of some eigenvalues.

The authors of this paper do not discuss the problem with loss of orthogonality and simply state they reorthogonalize. This is not adequate in this reviewer's opinion. Furthermore, the complexity computation does not consider the reorthogonalization work and therefore can be labeled as misleading. Despite the above this reviewer thinks that the effort to apply block Lanczos is important and there may be a way to achieve the complexity claimed by the authors, meaning not doing full reorthogonalization.

It could be valuable to see the author's experiments regarding the implementation when no reortogonalization is used. Did they run into the numerical issues prompting the reorthogonalization?

Section 2.2 the authors discuss stronger guarantees for low rank approximation. This is an interesting and valuable section with the introduction of per vector guarantee and the discussion of failures of prior low-rank approximation guarantees in practice.

The authors augment their theoretical work with a valuable experiment section, that shows the improved convergence. However, as mentioned above, using iteration count hides the costs of reorthogonalization and therefore more insightful discussion is warranted to discuss practical applicability.

The paper is well organized and the presentation is good. There are several typos and inconsistencies, listed below, that should be addressed by the authors to improve the menuscript.

38: Why mention rank r here at all? In addition k is used for rank later (see line 72). This can just cause slight confusion.

39: The authors (at least consistently) use a weird symbol to denote matrix transpose. Superscript T should work fine instead.

114: This reviewer expected K to have \Pi as well as shown on line 273.

119: This reviewer is confused why take the top k singular vectors of the k x k matrix M?

159: "instead of multiplicatively like" The reviewer understands that the per vector guarantee uses \sigma_{k+1} and not ]\sigma_i but what does multiplicatively has to do with this?

343, 373: There is more discussion need regarding reorthogonalization and its impact on computational cost.

424: This was published in 2010.

429: The author's first name should be: Tam\'as.

The line numbers below correspond to the full version:

507: Instead of just telling about the existence of p_1, the authors could give some motivation so that the reader knows somewhat why are we doing this. While this becomes clear later, it is good to give motivation to help the reader not to be puzzled.

524: It is quite unexpected and (not well defined in this reviewer's opinion) to use x < O(y). This reviewer understands that the authors try to convey that there is a constant when moving to the last inequality. Using big O in this manner is casual at best.

1070: This reviewer finds the formulation and proof a bit cumbersome. The use of \eta is not really necessary is it? One could say instead:

||A-B||_2^2 - ||A-A_k||_2^2 <= ||A-B||_F^2 - ||A-A_k||_F^2

The proof could be made more streamlined if continuing the above thought. This reviewer also suggests to recall (as it was mentioned before) that A_k is the best rank -k approximation and that ||A-A_k||_2^2 = \sigma_{k+1}^2, and similar for the Frobenius norm.

Additional comments following the author response and reviewer discussions.

I revised my review based on the discussions and the author responses. I change my rating to a 5.

However, I still think that with a major revision this could be a very valuable paper. My driving concern is that once the decision is made, there is no incentive to make the changes necessary. (At least that was my experience last year when reviewing.) So I am keeping my score as a slightly below acceptance. (If this were a journal publication, my call would recommend acceptance with a major revision.)

I implemented the algorithms and I believe that it has merit and could be useful in actual applications.

I went through the complexity computations. Please stay with me so I can make my point. Instead of using \epsilon I find it much more informative if we look at the cost in terms of q (number of iterations) and the cost of the matrix vector computation for A (and A') that I will denote by m. (m = 2*nnz(A) which I would expect to be about c*n assuming a constant number of non-zeros per column.)

Alg 1, Simultaneous iteration has cost breakdown: m(2*q+1)k

- to form K 2nk^2

- compute QR to get Q A'Q

- km form M

- dk^2 SVD of M

- 2k^3 form Z

- 2nk^2

Assuming k << q < n, the dominating term will be forming of K.

Alg 2, Block Lanczos has cost breakdown: m(2*q+1)k

- to form K (but also note much more memory needed) 2nq^2k^2

- compute QR to get Q A'Q

- qkm form M

- dq^2k^2 top k SVD of M - 2q^2k^3 form Z

- 2nqk^2

(Small note: forming M can be done by applying A' then A and then Q instead, the term is not dominating so we do not discuss which is better.)

The dominating term here appears to be the QR computation assuming m is about c*n.

The achieved accuracy is C_SI* log(d)/q and (C_BL*log(d)/q)^2, where C_SI, C_BL are constants for the algorithms respectively. This shows however that the accuracy achieved per computational effort is not satisfactory:

effort

accuracy (guarantee) SI

cn(2q+1)k

C_SI*log(d)/q BL

2nq^2k^2

(C_BL*log(d)/q)^2

Therefore it would seem that using SI with q^2 iterations is better. Of course the above ignores that in practice performance depends on use of back storage and in general memory etc. (But I am not the one submitting the paper.)

This however tells me that the approach to use QR to orthogonalize K is flawed. After all, the insight of the Lanczos method is in computing an orthogonal base efficiently. Here we use QR and not a recurrence. I for one even find the name Block Lanzos for this process misleading. It is a Krylov space based method, but no Lanczos recurrence about it as proposed. (However, this bad terminology could have gotten hold in the literature earlier and it is not the authors fault.)

I believe there is an efficient way of producing a Q that is "good enough" (in presence of numerical issues) that would work in practice, but more work is needed. I note that the idea has been out there before using the Krylov space, the trick is to make it work efficiently.

Some more comments below that would further improve the paper (in addition to the previous ones).

l. 115: Consider defining K with going up to q-1 only. As is written K is an n x (q+1)*k matrix. Otherwise, q downstream needs to be updated to (q+1) for correctness.

l 378: Compare the algorithms on a computational effort versus error plot as well as empirical time versus error plot. (The latter requires a faithfully good implementation of both algorithms, but not state-of-the art, just fair to the point being made.)

Emphasize that A is accessible as an operator through its action (matrix-vector product) in the targeted applications and that k is intended to be very small compared to n.

***

The implementation of the algorithms and the test script for reference:

SI.m: function [Z] = SI(A, k, q)

[n, d] = size(A);

P = randn(d, k);

B = A*A'; C = A*P; K = B*C; %keyboard;

[Q, R] = qr(K); Qk = Q(:, 1:k);

M = Qk'*B*Qk;

[U, S, V] = svd(M);

Uk = U(:, 1:k);

Z = Qk*Uk; return;

**** BL.m: function [Z] = BL(A, k, q)

[n, d] = size(A);

P = randn(d, k);

B = A*A'; C = A*P; K = zeros(n, q*k); for i = 0:q-1

K(:, i*k + 1:(i+1)*k) = C;

C = B*C; end

[Q, R] = qr(K); Qk = Q(:, 1:q*k);

M = Qk'*B*Qk;

[U, S, V] = svd(M);

Uk = U(:, 1:k);

Z = Qk*Uk; return;

test.m: *** rand("state", 423);

n = 1000; d = 300; k = 5; tol = 1e-3; q = log(d)/sqrt(tol)*0.01 q = 10;

A = randn(n, d); [U, S, V] = svd(A); Uk = U(:, 1:k); Ak = Uk*Uk'*A;

Z1 = SI(A, k, q); Z2 = BL(A, k, q);

norm(A - Z1*Z1'*A)/norm(A - Ak) norm(A - Z1*Z1'*A, 'fro')/norm(A - Ak, 'fro')

norm(A - Z2*Z2'*A)/norm(A - Ak) norm(A - Z2*Z2'*A, 'fro')/norm(A - Ak, 'fro')
Summary: The paper is well written and addresses an important problem. However a few concerns (see in detail below) prevent this reviewer to recommend the paper for acceptance. Should these concerns be adequately addressed by the authors, the paper would reach the threshold of acceptance and in fact could be a valuable and interesting paper.

Submitted by Assigned_Reviewer_3

Quality and originality

This paper propose to solve the truncated SVD by a Krylov subspace method and establish strong convergence guarantees. This paper contributes the first gap-independent bound of the block Lanczos method. This result is novel.

Significance

The block Lanczos is a classical and extensively applied algorithm in numerical linear algebra. Improving the error bound of such a decades-old algorithm is not easy and should be considered as a great contribution.

-----

Clarity

This paper is very well written. But I have some suggestions:

1. Considering that the Lanczos methods are decades old, and many convergence analyses have been established. So the authors should explain the existing convergence bounds, especially the prior art, in which the readers are interested. At least the bound of Saad should be discussed.

2. Traditionally, the block Lanczso method sets the block size far smaller than $k$. So the authors should discuss the reasons why they set the block size so big.

----

Correctness

* I hope the authors can answer the following question

In Algorithm 2, the matrix monomials (the block of K) have orders 1, 3, 5, 7, ... However, the matrix polynomial p1(A) is the sum of matrix monomials of orders 0, 1, 2, 3, ..., q. Apparently the even-order matrix monomials are missing. So I am wondering why the range of p1(A) is contained in the Krylov subspace. The argument in Line 513 does not make sense because (A A')^{(i-1)/2} is not evaluated. It appears that A p1(A A') should be used in place of p1(A).

I am expecting a rigorous proof of "$Range (p1(A) \Pi) \subset Range(K)$", with every step well explained. The authors are encouraged to provide me a link of Google Drive, Dropbox, or Skydrive.

* Errors which can be fixed

- The third property in Lemma 5 should be |p(x)| < ... Otherwise, the inequality in Line 524 does not hold.

E.g. -10 < 1 does not imply (-10)^2 < 1^2

- In the proof of Lemma 5 Eq 15: should restrict x > 1? last line in page 19: \epsilon should be \gamma? Due to Eq 16 and Eq 20, should property 3 be ...2^{2 q \sqrt{\gamma} - 2} rather than 2^{2 q \sqrt{\gamma} - 1}?
Summary: This paper improves the error bound of an extremely important algorithm.

I have carefully checked the proof and spotted several errors, one of which I do not know how to fix. I am expecting the authors to provide additional proof.

------ reply to the authors' response:

1. Now I think the proof is correct. The authors should add $range(p1(A)X) \subset range (K)$ as a lemma to the appendix. It is not obvious that odd order Chebyshev polynomial is the sum of odd order monomials.

2. From their feedback, I find that the authors may still have misunstanding of the block Lanczos method. It is unnecessary to set the block size greater than $k$, and small block size has nothing to do with deflation. In fact, with the small block size $b = r - k + 1$, virtually the same gap-dependent bound has been established in the liturature. I strongly suggest that the authors should look into the literature and discuss the effects of block size. I think big block size should have certain positive effects, but you must convince people from the numerical linear algebra society of this point.

Submitted by Assigned_Reviewer_4

Summary:

In this paper authors analyze block Lanczos method (Krylov subspace technique) in the context of approximate/randomized SVD. They consider alogrithms with power iteration technique (Rokhlin, Szlam, Tygert) to increase the gap between k and k+1 singular values. They show that different from standard simultaneous iteration algorithm (Halko, Martinsson, Tropp) Lanczos method needs only O(1/sqrt(epsilon)) power iterations instead of O(1/epsilon), to give an epsilon spectral norm approximation guarantee. Interestingly authors show that these algorithms satisfy stronger per vector error guarantees further reducing the gap between the theory and performance observed in practice.

Comments:

1. Classical power method (orthogonal iterations) for computing rank-k approximation has a complexity of log(1/epsilon) / log(lambda_k/lmabda_{k+1}) iterations. Simultaneous iteration and block Lanczos randomized techniques need atleast O(log(n)/sqrt(epsilon)) power iterations indepedent of the spectral gap. Apriori it is not clear that these randomized techniques are better in all regimes, since they need much more passes over the data than classical power method and don't have lesser computation like sketch-and-solve methods. A

comparison with the classical techniques (power method and RRQR http://epubs.siam.org/doi/abs/10.1137/0917055) is lacking both in theory and in experiments. A discussion or a simple experiment showing clearly the regime where these techniques are better will be helpful.

2. While block Lanczos method has faster convergence than simulataneous iteration, it needs q times more memory. Authors have mentioned a possible way to avoid it in section 5.2, though it is not clear if it reduces memory usage. A comment on this will be helpful.

3. http://epubs.siam.org/doi/abs/10.1137/130938700 This paper seems to have some new results for problems considered.
Summary: The guarantees for block Lanczos method for randomized SVD setting and the new per vector error analysis seems very interesting. Overall the improvements are substantial and easily noticed in practice. However these algorithms need too many passes over the data than simple sketch-and-solve methods and a discussion comparing this with classical methods is missing.

Author Feedback
Author rebuttal: First, we want to thank the reviewers for many helpful comments and suggestions.

Several asked us to expand comparison with classical bounds, including those of Saad and bounds for non-block methods with random starts. We will be sure to do so. The randomized algorithms studied are really classical methods with randomized initializations so, in addition to our new gap independent bounds, they match prior work. In fact, our gap dependent analysis (Theorem 13 of the supplemental) can be used to recover many classical results.

Review 2:

You're right - a discussion of why we use block size k would be valuable. Essentially, without costly (and potentially unstable) deflation operations, a block size of r is required to recover r singular vectors with the same (or nearly the same) value (see e.g. Golub & Van Loan, Sec. 9.2). In the extreme, all top singular values could be close so, for gap independent bounds, we need to set r = k.

As requested, we posted a formal proof that range(p1(A)Pi) \subset range(K) at: http://document.li/NsV6. In short, if we choose p1(A) to be a scaled Chebyshev polynomial of *odd* degree then it only includes monomials of odd degrees 1, 3, 5, etc.

Thank you for pointing out other errors.

Review 3:

1. We will expand our comparison with classical methods - sorry for the confusion. Randomized Simultaneous Iteration actually matches the cited iteration count of log(1/epsilon)/log(lambda_k/lambda_{k+1}. This is proven in Theorem 13 of the supplemental. If we set p = k, our gap dependence is 1/(lambda_k/lambda_{k+1}-1) < 1/log(lambda_k/lambda_{k+1}) since, using the natural log, 1/(x-1) < 1/log(x). Theorem 13 also shows that Block Lanczos beats this.

In classical analysis it is (implicitly) assumed that we start with initial vectors that have constant dot product with the true top singular vectors. Without this assumption, log(d) additional iterations are required, which is why we get a log(d/epsilon) = log(d) + log(1/epsilon) dependence. If we assumed constant starting dot products the dependence would match the classical log(1/epsilon).

So, our analysis always at least matches the classical bounds. At the same time, when lambda_k/lambda_{k+1} is close to 1 (and hence 1/log(lambda_k/lambda_{k+1}) is very large), our gap independent bounds can be significantly stronger. See the brief discussion in Sec. 6. Again, thanks to your suggestion, this will all be expanded upon.

Like sketch-and-solve methods, RRQR can only give a Frobenius norm guarantee, not the stronger spectral norm or approximate PCA guarantees achieved by the iterative algorithms. As emphasized in Figure 5 for sketch-and-solve, this can translate to poor practical performance. A comparable experiment for RRQR would be valuable and we will add.

2. This is a good point. Even with the faster implementation mentioned in 5.2, Block Lanczos requires more memory than Simultaneous Iteration. We will add a comment.

Review 5:

We will clarify the treatment of reorthogonalization/stability. The issues you point out largely arise when a recurrence is used to obtain a (block) tridiagonal form for Q^TAA^TQ. This is essential if you want to compute many singular vectors since your post-processing step only needs to compute the SVD of a block tridiagonal matrix.

On the other hand, we compute the Krylov subspace directly and post-processes it with dense orthogonalization/SVD operations. For learning applications, where k is small, this is standard - see our referenced implementations of randomized Block Lanczos. The cost of the SVD is dominated by the cost of computing the Krylov subspace especially when data needs to be loaded from disk on each iteration, while post-processing can typically happen in main memory.

Without the recurrence, stability issues become less delicate. We reorthogonalize our iterate block so that it does not become ill-conditioned, which is standard for block power methods (see "Normalized Power Iterations for the Computation of SVD" by Martinsson, Szlam, & Tygert or Sec. 7.3.2 of Golub & Van Loan). Note that this does not require reorthogonalizing against prior blocks of the Krylov subspace and thus the runtime of our implementation fits within the claimed complexity of Theorem 8. We apologize that this was misleading and will clarify. Specifically, the cost of orthogonalizing is O(nk^2*num_blocks) = O(nk^2*logd/\sqrt{epsilon}), which is less than the cost of performing the final SVD.

"why take the top k singular vectors of the k x k matrix M?": For Simultaneous Iteration, this is necessary for per vector error (equation 3). If we are only interested in low rank approximation, we don't need to compute M and can just set Z = Q. For Block Lanczos, the Krylov subspace has more than k columns - taking the SVD ensures we grab the directions of greatest variance (see lines 275-277)

Thank you very much for numerous other errors found and suggested improvements.